# Epidermal RAF prevents allergic skin disease

**Josipa Raguz[1], Ines Jeric[1], Theodora Niault[1], Joanna Daniela Nowacka[1], Sanya Eduarda Kuzet[1], Christian Rupp[1], Irmgard Fischer[1], Silvia Biggi[2], Tiziana Borsello[2,3], Manuela Baccarini[1]***

[1]Department of Microbiology, Immunology and Genetics, Max F. Perutz Laboratories, University of Vienna, Vienna, Austria; [2]Department of Neuroscience, IRCCS Istituto di Ricerche Farmacologiche "Mario Negri", Milano, Italy; [3]Department of Pharmacological and Biomolecular Sciences, Università degli Studi di Milano, Milano, Italy

**Abstract** The RAS pathway is central to epidermal homeostasis, and its activation in tumors or in Rasopathies correlates with hyperproliferation. Downstream of RAS, RAF kinases are actionable targets regulating keratinocyte turnover; however, chemical RAF inhibitors paradoxically activate the pathway, promoting epidermal proliferation. We generated mice with compound epidermis-restricted BRAF/RAF1 ablation. In these animals, transient barrier defects and production of chemokines and Th2-type cytokines by keratinocytes cause a disease akin to human atopic dermatitis, characterized by IgE responses and local and systemic inflammation. Mechanistically, BRAF and RAF1 operate independently to balance MAPK signaling: BRAF promotes ERK activation, while RAF1 dims stress kinase activation. In vivo, JNK inhibition prevents disease onset, while MEK/ERK inhibition in mice lacking epidermal RAF1 phenocopies it. These results support a primary role of keratinocytes in the pathogenesis of atopic dermatitis, and the animals lacking BRAF and RAF1 in the epidermis represent a useful model for this disease.

*For correspondence: manuela.baccarini@univie.ac.at

**Competing interests:** The authors declare that no competing interests exist.

## Introduction

The largest organ of the body, the skin, allows exchange with the environment while shielding the organism from insults of mechanical, chemical, and infectious nature. In the skin, the epidermis acts as a mechanical barrier which prevents water loss and the entry of potentially harmful chemicals; in addition, through the interplay between keratinocytes in the epidermis and immune cells based mainly in the underlying dermis, the skin works as an immunological barrier actively defending the body from pathogens (*Pasparakis et al., 2014*; *Nestle et al., 2009*). Maintaining these barrier functions throughout life requires continuous regeneration of the epidermis and appropriately balanced immune responses. Dysregulation of the regenerative process can lead to a wide range of defects, from epidermal thickening to malignancies; and an imbalance in the skin's immune reactions can lead to recurrent infections, or inflammatory/allergic diseases such as psoriasis or atopic dermatitis.

The RAF/MEK/ERK signaling pathway plays an essential role in the epidermis. Downstream of the EGFR and RAS, it acts in hair follicle development and wound healing; its mild activation in Rasopathies, genetic diseases caused by activating mutations in the pathway, results in skin phenotypes ranging from thickening of palms and soles to the development of papillomas; and strong activation in keratinocytes results in tumorigenesis (*Kern et al., 2011*; *Ratushny et al., 2012*). In animal models, epidermis-restricted inducible activation of RAF or MEK causes massive cutaneous hyperplasia and reduced differentiation (*Khavari and Rinn, 2007*). Consistently, RAF1 (also known as CRAF) and BRAF are essential for the development and progression of RAS-induced tumors although they fulfil

**eLife digest** The skin is the largest organ of the body and shields it from damage. It is also home to cells of the immune system, which protect the body from infections. To maintain its role as a barrier, the skin regenerates throughout life, constantly producing new cells to replace the old ones. If this process goes wrong and the skin regenerates too much, the skin may produce tumors. Likewise, disrupting the immune barrier can lead to autoimmune or allergic skin diseases such as psoriasis or atopic dermatitis (also known as eczema).

The outer layer of the skin is called the epidermis, and is made up of cells known as keratinocytes. A family of proteins called RAF plays an important role in controlling how keratinocytes behave. These proteins are part of a signaling pathway that controls the production of new cells and is disrupted in skin tumors. Therapies that inhibit RAF are effective treatments for these tumors but have side effects that can affect the skin so severely that the treatment must be interrupted.

Keratinocytes also play a role in the development of allergic skin diseases. However, it is not known whether they do so by triggering the disease themselves, or by responding to stimuli produced by immune cells.

Raguz et al. investigated what would happen if RAF proteins were removed from keratinocytes in the epidermis of mice. This caused the mice to develop an allergic disease similar to human atopic dermatitis. This was an unexpected effect and different from the side effects caused by drugs that inhibit RAF proteins. By analyzing the signaling pathway that RAF is part of, Raguz et al. discovered that removing RAF from the epidermis reduces the pathway's ability to prevent excessive stress signals being sent throughout the skin. Under these conditions, the keratinocytes bring about inflammation and allergy by activating the immune cells in the skin and elsewhere.

Overall, the results presented by Raguz et al. indicate that allergic dermatitis can be triggered by defects in keratinocytes rather than in immune cells. Furthermore, RAF in the epidermis appears to prevent allergic skin diseases. Future studies could use mice that lack RAF in their epidermis to further understand atopic dermatitis and to investigate the way in which drugs that target RAF can damage the skin.

different roles, with RAF1 preventing keratinocyte differentiation (*Ehrenreiter et al., 2009*) and BRAF promoting proliferation through MEK/ERK activation (*Kern et al., 2013*). The requirement for MEK/ERK in epidermal proliferation has been independently assessed (*Khavari and Rinn, 2007*; *Dumesic et al., 2009*; *Scholl et al., 2009*; *Scholl et al., 2009*).

In line with their prominent roles in cell proliferation and tumorigenesis in a number of tissues, the components of the EGFR/RAS/RAF/MEK/ERK pathway are attractive targets for molecule-based therapy. Many inhibitory compounds have been developed that are currently in clinical trials or have reached the clinic. Reflecting the function of the pathway in skin, cutaneous toxicities are one of the main adverse effects of these therapies; they can be severe and lead to an interruption of the therapy or to its termination (*Belum et al., 2013*; *Curry et al., 2014*; *Dy and Adjei, 2013*). These adverse effects can be roughly classified as inflammatory reactions, elicited chiefly by blocking EGFR or MEK; and proliferative events, caused by multikinase inhibitors such as sorafenib or by more specific RAF inhibitors (vemurafenib, dabrafenib). Cutaneous inflammation induced by agents blocking EGFR (erlotinib or cetuximab) and by MEK inhibitors (selumetinib and trametinib) is characterized by papulopustular rash, pruritus, and suppurative folliculitis in $\geq$80% of the patients (*Curry et al., 2014*). The effects of erlotinib and cetuximab are on-target, as recently demonstrated by two studies showing similar inflammatory phenotypes in mice with epidermis-restricted EGFR ablation (*Mascia et al., 2013*; *Lichtenberger et al., 2013*).

Skin rashes and pruritus are also observed in patients treated with inhibitors targeting RAF. Up to half of the cutaneous reactions induced by these compounds, however, consist of anomalous epidermal proliferation events ranging from different forms of keratosis to the development of drug-induced papillomas, keratoacanthomas, and squamous cell carcinomas (4–30% depending on the study and the inhibitors used) (*Anforth et al., 2013*). These side effects are mechanism-based and

rely on the paradoxical activation of MEK/ERK promoted by the inhibitors in cultured cells, animal models, and patients (*Samatar and Poulikakos, 2014*; *Holderfield et al., 2014*). Consistent with this, combination treatment (RAF plus MEK inhibitors) reduces cutaneous toxicity (*Flaherty et al., 2012*; *Mattei et al., 2013*).

In contrast to RAF inhibitor treatment, inducible epidermis-restricted ablation of BRAF and RAF1 causes the rapid regression of RAS-driven tumors without apparent cutaneous toxicity (*Kern et al., 2013*). Here, we systematically test the effect of compound BRAF/RAF1 ablation in epidermis and show that it gives rise to a progressive disease strongly resembling human atopic dermatitis. Mechanistically, the disease relies on the combination of reduced ERK and increased stress kinase activation, leading to chemokine/cytokine overproduction and chronic, systemic inflammation.

## Results

### Epidermal BRAF and RAF1 are essential to prevent local and systemic inflammation

BRAF and RAF1 were ablated in the epidermis by introducing the *Krt5-Cre* transgene (*Tarutani et al., 1997*) into a *Braf* $^{f/f}$; *Raf1*$^{f/f}$ background (*Kern et al., 2013*) (*Figure 1A*). The mice (heretofore referred to as Δ/Δep2) were born at Mendelian ratio but with their eyes open (*Figure 1B*), probably as a result of the migration defects of RAF1-deficient keratinocytes (*Ehrenreiter et al., 2005*). With time, however, they started to show symptoms of a progressive skin disease. These included intense itching and scratching resulting in partial alopecia and self-inflicted wounds (*Figure 1B*). Histological examination of the non-affected areas revealed thickening of the epidermis correlated with increased proliferation and expansion of both the basal and suprabasal keratinocytes but not with keratinocyte apoptosis (*Figure 1C* and *Figure 1—figure supplement 1A*). Filaggrin (FLG) expression was similar to that of the controls (*Figure 1—figure supplement 1A*). We also observed a rich dermal infiltrate comprised of activated (β1 Tryptase+) mast cells, granulocytes, dendritic cells and, less abundant, T cells and macrophages (*Figure 1C–D* and *Figure 1—figure supplement 1B*). In line with the dermal inflammatory reaction, Δ/Δep2 epidermis was characterized by the robust expression of the keratinocyte activation marker K6, of the cell adhesion molecule ICAM1, and by the sporadic expression of MHC class II molecules (*Figure 1E*), all upregulated in inflammatory conditions including atopic dermatitis (*Freedberg et al., 2001*; *Fan et al., 2003*; *Caughman et al., 1992*). Consistent with this, Δ/Δep2 epidermal lysates enriched in K5/K10-positive keratinocytes (*Doma et al., 2013*) contained increased amounts of the cytokine TSLP, associated with skin allergic disorders (*Ziegler et al., 2013*) (*Figure 1F*), as well as of other chemokines and cytokines, with CCL7, IL18, IL5 and IL13 levels significantly higher than controls (*Figure 1G*).

Epidermal ablation of BRAF and RAF1 had profound systemic effects. The mice failed to thrive (*Figure 2A*) and presented with enlarged spleens and lymph nodes. The splenomegaly could largely be attributed to increased numbers of Mac1+Gr1+ cells (*Figure 2B*), a cell type found both in the spleen of a mouse model of FITC contact hypersensitivity/*S. aureus* infection and in the blood and skin infiltrates of atopic dermatitis patients (*Skabytska et al., 2014*). The lymph nodes contained elevated numbers of activated T, B, and dendritic cells (*Figure 2C*). In the blood, we observed leukocytosis and increased amounts of the chemokines CCL2 and CCL7, as well as of GCSF; in addition, serum IgEs were elevated in Δ/Δep2 mice (*Figure 2D*). Thus, the phenotype of adult Δ/Δep2 mice resembled a skin-specific allergic disease.

To determine whether the systemic phenotype was secondary to the severe skin inflammation observed in adult animals, we examined Δ/Δep2 mice at weaning (3 weeks of age), at which stage they did not groom more than control littermates nor showed any signs of skin rash, except a mild eyelid irritation (*Figure 3—figure supplement 1A*). Skin architecture and keratinocyte proliferation were not altered at this stage (*Figure 3—figure supplement 1A*). In terms of dermal infiltrate, a 2-fold increase in mast cells could already be observed, but these cells were not activated (*Figure 3A* and *Figure 3—figure supplement 1A*). Granulocytes and dendritic cell numbers were indistinguishable in 3 weeks old F/F2 and Δ/Δep2 littermates (*Figure 3—figure supplement 1A*). In the epidermis, ICAM1 expression was slightly upregulated, while K6 and MHC II expression could not be detected (*Figure 3A* and *Figure 3—figure supplement 1B*). TSLP, CCL2, CCL7, and a number of cytokines (GMCSF, IL6, 4, 5, 13, 2) were already significantly elevated in the epidermal lysates, while

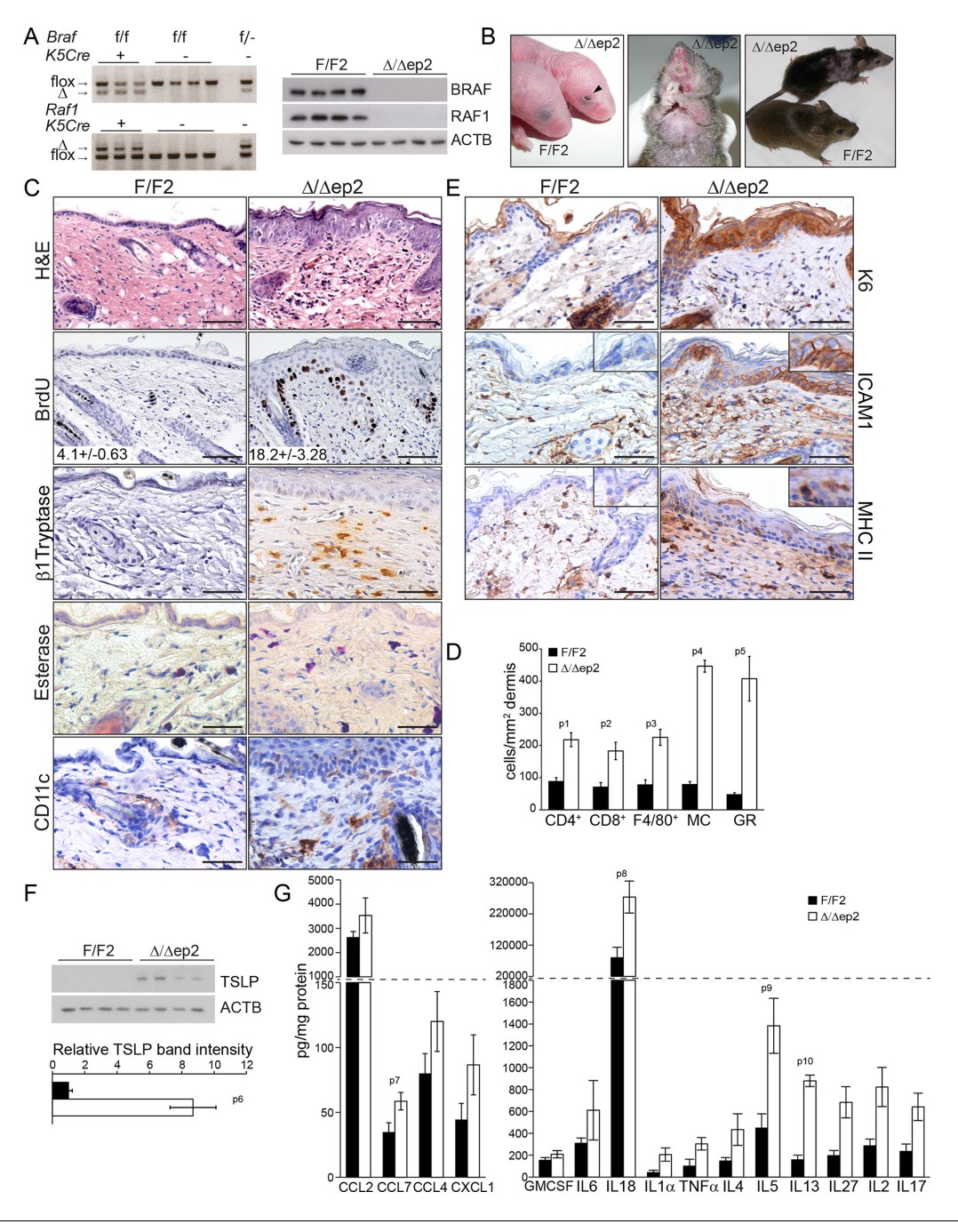

**Figure 1.** Compound deletion of BRAF and RAF1 in the epidermis leads to severe skin inflammation in adult mice. (**A**) BRAF and RAF1 are efficiently deleted in epidermal cells as shown by PCR analysis of tail tissue and immunoblotting of epidermal lysates isolated from 3 weeks old F/F2 and Δ/Δep2 (n = 4). ACTB is shown as a loading control. (**B**) Macroscopic appearance of newborn and adult F/F2 and Δ/Δep2 mice. Arrowhead highlights the open eye phenotype of Δ/Δep2 pups. (**C**) Hematoxylin/eosin (H and E) staining shows epidermal thickening and dermal inflammatory infiltrates in Δ/Δep2 mice. BrdU incorporation confirms hyperproliferation in the basal layer of Δ/Δep2 epidermis. The numbers in the inset represent BrdU$^+$ cells/mm$^2$ of epidermis (n = 5–7, mean ± SEM). Infiltrating cells: activated mast cells (β1 Tryptase$^+$), granulocytes (esterase$^+$), dendritic cells (CD11c$^+$). (**D**) Quantification of the infiltrating cells: T cells (CD4$^+$ and CD8$^+$), macrophages (F4/80$^+$), total mast cells (MC, toluidine blue$^+$), granulocytes (GR, esterase$^+$). (**E**) Increased expression of K6, ICAM1, and MHC II in Δ/Δep2 keratinocytes/epidermis. Representative images (**C, E**) and quantification (**D**) of 5–7 individual couples. Scale bars, 50 μm. (**F**) Inflammatory chemokines and cytokines in epidermal lysates (n = 3–4). TSLP levels were determined by

*Figure 1 continued on next page*

*Figure 1 continued*

immunoblotting and quantified by Image J. ACTB served as a loading control. The results were normalized by arbitrarily setting one of the F/F2 samples as 1 and plotted as mean ± SEM. Data represent mean ± SEM. p = 0.011, p1 = 0.001, p2 = 0.007, p3 = 0.001, p4 = 3.06E-6, p5 = 1.37E-4, p6 = 0.002, p7 = 0.049, p8 = 0.042, p9 = 0.046 and p10 = 0.001.

The following figure supplement is available for figure 1:

**Figure supplement 1.** Local inflammation in adult mice lacking BRAF and RAF1 in the epidermis.

---

only a trend could be observed for others (*Figure 3B* and *Figure 3—figure supplement 1C*). Systemically, enlarged lymph nodes contained increased amounts of activated T, B, and dendritic cells, while the spleen was normal both in size and cell composition (*Figure 3C* and *Figure 3—figure supplement 1D*). In the blood, the number of monocytes and granulocytes was increased, as were the levels of CCL2, CCL7, and GCSF. However, no significant difference in serum IgEs could be observed at this point (*Figure 3D*). The increase in T, B, and dendritic cells in lymph nodes and the granulocytosis could be traced as far back as 10 days of age, at which time point GCSF was the only cytokine elevated in the serum of Δ/Δep2 mice (*Figure 3E–F*). Thus, ablation of both *Braf* and *Raf1* in mouse epidermis does not directly disturb skin architecture but rather affects the cross-talk between keratinocytes and the innate and adaptive immune system.

## Lack of epidermal BRAF and RAF1 causes transient inside-out barrier defects

Inflammatory skin conditions often involve impaired skin barrier function. Δ/Δep2 embryos performed normally in an outside-in dye penetration assay, indicating that stratum corneum permeability was not affected (*Figure 4A*). We next monitored the weight of E18.5 embryos of different

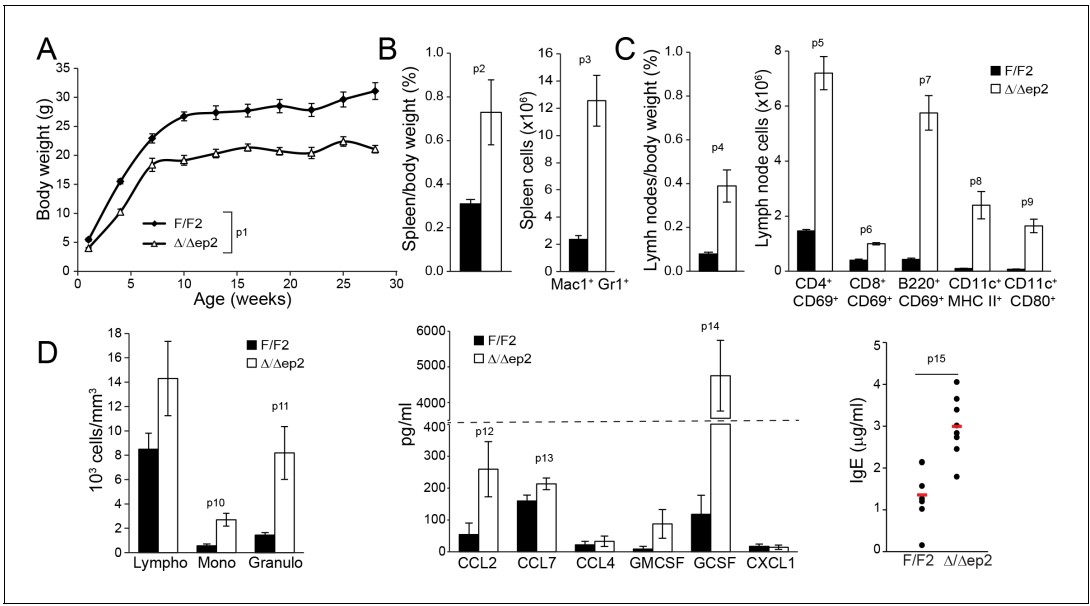

**Figure 2.** Inflammatory response in adult Δ/Δep2 animals. (**A**) The body weight of Δ/Δep2 mice is significantly reduced compared to their littermates (n = 5). The data was analyzed by two-way analysis of variance (ANOVA) test. (**B**) Increased spleen/body weight ratio and increased numbers of splenic Mac1+Gr1+ cells in adult Δ/Δep2 animals (n = 4–5). (**C**) Enlarged lymph nodes and and activated T, B, and dendritic cells in adult Δ/Δep2 (n = 4–8). T cells (CD4+ or CD8+) and B cells (B220+) activation was determined by costaining with CD69; activated dendritic cells were identified as CD11c+ and MHC IIhi or CD80+. (**D**) Circulating blood cells and plasma levels of chemokines and IgE antibodies in adult mice (n = 6–8). Data represent mean ± SEM. p1 = 0.0002, p2 = 0.023, p3 = 0.002, p4 = 0.002, p5 = 9.83E-8, p6 = 6.31E-7, p7 = 2.91E-7, p8 = 0.004, p9 = 0.001, p10 = 0.008, p11 = 0.011, p12 = 0.046, p13 = 0.050, p14 = 3.65E-4 and p15 = 0.001.

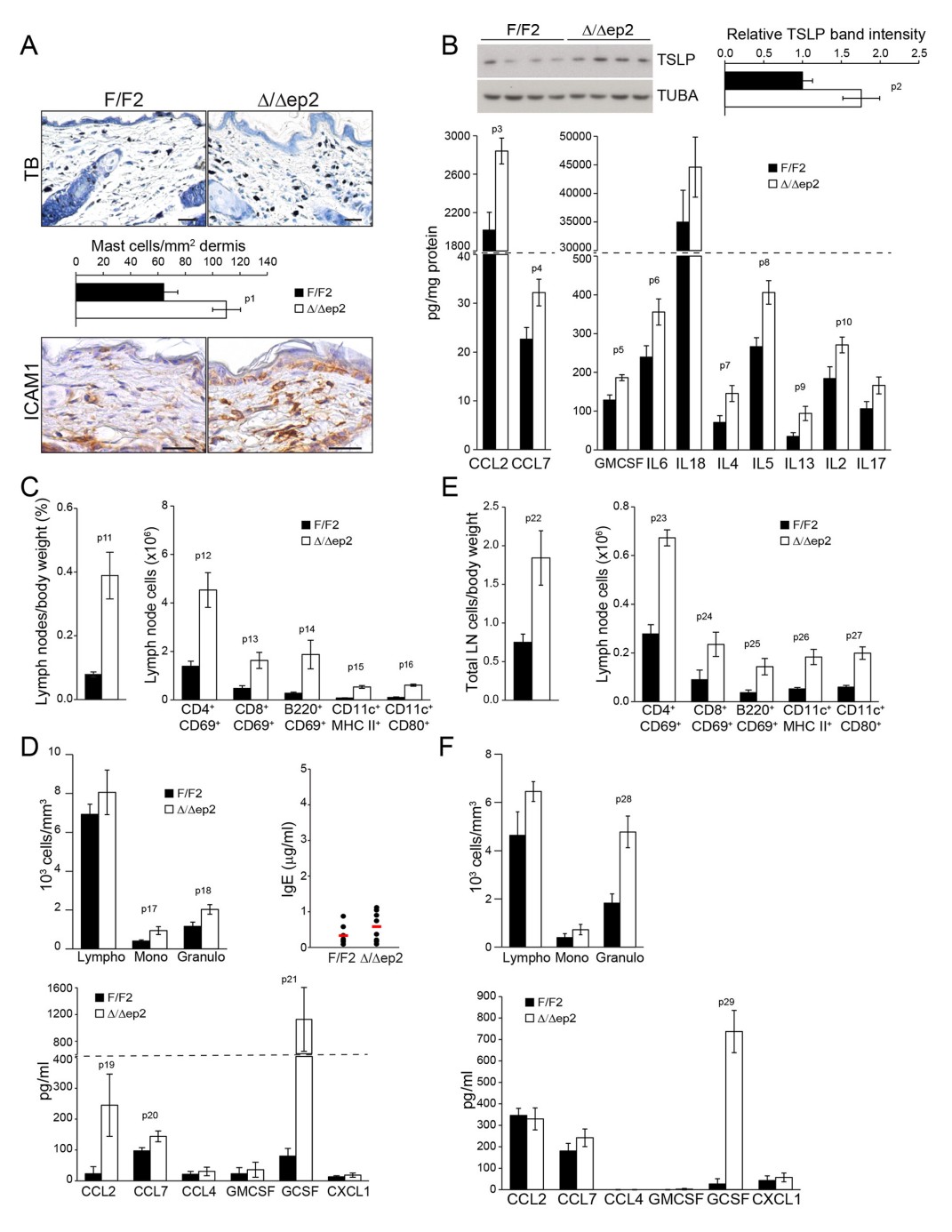

**Figure 3.** Local and systemic inflammatory phenotype in young Δ/Δep2 animals. (**A**) Local inflammation in 3 weeks old Δ/Δep2 animals. Total mast cells (toluidine blue staining, TB; quantified in the plot, n = 4–5) and ICAM1 staining. Scale bars, 25 μm. (**B**) Inflammatory chemokines and cytokines in epidermal lysates (n = 4–5). TSLP levels were determined by immunoblotting and quantified and analyzed as in *Figure 1F*. TUBA served as a loading control. (**C, D**) Systemic inflammatory parameters in 3 weeks old mice. (**C**) Lymph node size and composition (n = 4). (**D**) Circulating blood cells (n = 8) and plasma concentration of IgE (n = 8) and chemokines (n = 9). (**E, F**) Systemic inflammatory parameters in 10 days old mice. (**E**) Lymph node size and composition (n = 4–10). (**F**) Hemogram showing elevated amounts of granulocytes (upper panel, n = 4–7) and plasma chemokine levels showing increased GCSF (n = 4). Data represent mean ± SEM. p1 = 0.016, p2 = 0.041, p3 = 0.013, p4 = 0.040, p5 = 0.013, p6 = 0.032, p7 = 0.026, p8 = 0.007, p9 = 0.029, p10 = 0.048, p11 = 0.015, p12 = 0.018, p13 = 0.033,

*Figure 3 continued on next page*

*Figure 3 continued*

p14 = 0.036, p15 = 3.00E-04, p16 = 3.88E-05, p17 = 0.026, p18 = 0.021, p19 = 0.048, p20 = 0.034, p21 = 0.042, p22 = 0.008, p23 = 0.001, p24 = 0.018, p25 = 0.005, p26 = 0.001, p27 = 0.001, p28 = 0.011 and p29 = 0.014.

The following figure supplement is available for figure 3:

**Figure supplement 1.** Local and systemic response in 3 weeks old △/△ep2 animals.

genotypes for 9 hr after birth as a proxy for transepidermal water loss and thus for the barrier function of tight junctions. Δ/Δep2 embryos lost three-fold the weight of littermate controls (*Figure 4B*). Molecularly, reduced expression of E-cadherin (CDH1), an adherens junction protein known to regulate tight junctions (*Tunggal et al., 2005*), and of the tight junction protein claudin 1 (CLDN1), which is crucial for the maintenance of the inside-outside barrier (*Furuse et al., 2002*) and has been found downregulated in atopic dermatitis patients (*De Benedetto et al., 2011*) was still evident at P3 in Δ/Δep2 epidermis, while occludin expression was normal (OCLN) (*Figure 4C*). The architecture of P3 Δ/Δep2 skin was indistinguishable from that of controls (*Figure 4—figure supplement 1A*); P3 epidermal lysates contained highly variable amounts of chemokines and cytokines, predominantly CCL2, IL6 and IL18. All were increased in the Δ/Δep2 epidermal lysates, albeit not significantly (*Figure 4—figure supplement 1B*). The E-cadherin and claudin 1 downregulation was transient and was no longer detectable in 3 weeks old Δ/Δep2 mice (*Figure 4D*). Single deletion of RAF1 (RAF1Δ/Δep) and BRAF (BRAFΔ/Δep) did not lead to increased body weight loss in E18.5 embryos; consistently,

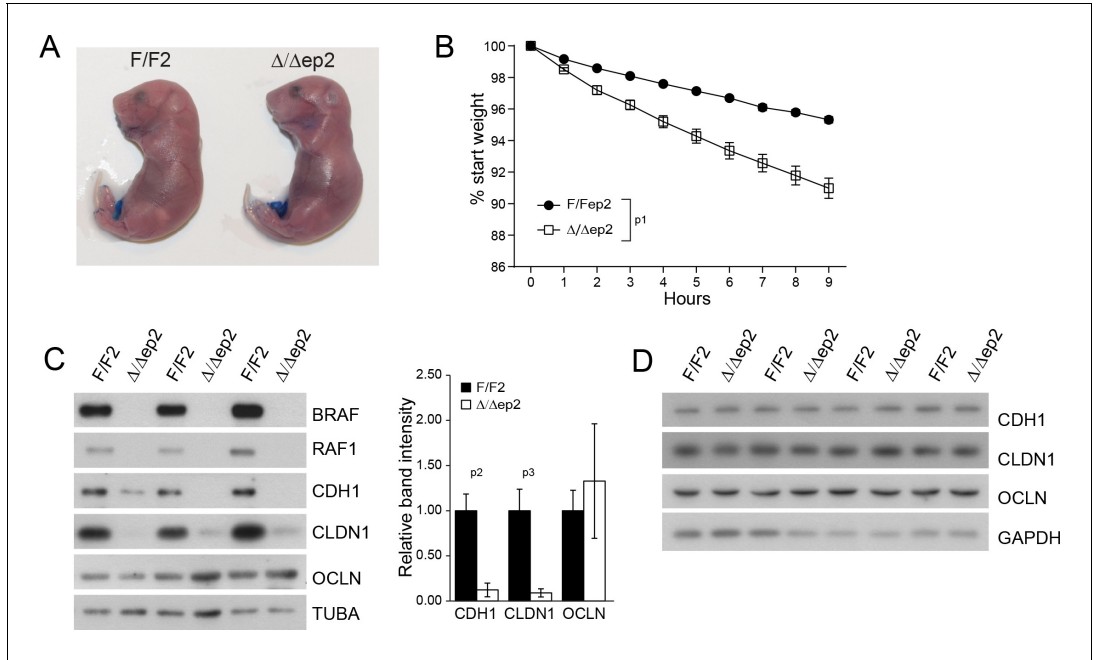

**Figure 4.** Transient inside-outside barrier defects in Δ/Δep2 animals. (A) Intact outside-in barrier function (determined by toluidine blue penetration of the stratum corneum) in E19.5-day-old Δ/Δep2 embryos compared to controls (n = 6). Representative pictures; two independent experiments were performed with identical results. (B) Increased water loss in the Δ/Δep2 E18.5 embryos as demonstrated by weight analysis. Results are displayed as percentage of initial weight (n = 41 for F/F2 and n = 11 for Δ/Δep2). The data was analyzed by two-way analysis of variance (ANOVA) test. (C, D) Immunoblot analysis of CDH1, CLDN1 and OCLN expression in epidermal lysates of 3 days old (C, n = 3; quantification shown in the plot, performed as in *Figure 1F*) or 3 weeks old Δ/Δep2 animals. TUBA and GAPDH are shown as loading controls. Data represent mean ± SEM. p1 = 0.0001, p2 = 0.028, p3 = 0.020.

The following figure supplement is available for figure 4:

**Figure supplement 1.** Skin architecture and inflammatory factors in 3 days old F/F2 and △/△ep2 animals.

the expression levels of tight junction proteins were normal in P3 epidermis (*Figure 4—figure supplement 1C–D*).

To assess the relevance of the transient perinatal barrier defect of Δ/Δep2 mice, we deleted *Braf* and *Raf1* in 3 weeks old mice using tamoxifen-inducible Krt5-Cre (*Indra et al., 1999*). These animals, termed Δ/Δep2$^{TX}$, showed conversion of both F to Δ alleles and strongly reduced expression of BRAF and RAF1 proteins in tail tissue (*Figure 5A*). Δ/Δep2$^{TX}$ mice developed a milder disease than the Δ/Δep2 mice, characterized by much slower kinetics (8 months between ablation and overt symptoms), moderate keratinocyte hyperproliferation (assessed as increased epidermal thickness) and activation (as determined by K6 and ICAM1 expression), as well as by a modest increase in activated mast cells and granulocytes in the dermis (*Figure 5B*). At the systemic level, we observed mild splenomegaly with an increase in Mac1+Gr1+ cells (*Figure 5C*) as well as enlarged lymph nodes containing activated lymphocytes and dendritic cells (*Figure 5D* and *Figure 5—figure supplement 1*). Increased numbers of lymphocytes and granulocytes were found in the blood, while IgE levels were comparable to controls (*Figure 5E*). Thus, circumventing the transient barrier defect of the Δ/Δep2 animals postponed and attenuated the clinical manifestation of the disease.

## Increased JNK and reduced ERK activation in Δ/Δep2 epidermis and cells

To gain insight in the molecular correlates of the phenotype, we analyzed MAPK signaling in the epidermis. We observed decreased activation of ERK and increased activation of JNK, assessed by the phosphorylation of both MAPK and their respective downstream targets pRSK and pJUN, in non-lesional epidermis of adult Δ/Δep2 and Δ/Δep2$^{TX}$ mice; in contrast, phosphorylation of the p38 target MAPKAPK2 was not affected (*Figure 6A and B*). A more detailed analysis of 3 weeks old animals confirmed low ERK activation and increased phosphorylation of JNK (and, to a lower degree, p38) in Δ/Δep2 epidermal lysates (*Figure 6C*). Consistent with this phosphorylation pattern, the levels of the dual specificity phosphatase DUSP1 and DUSP10, negative regulators of the stress kinases (*Patterson et al., 2009*), were low in the Δ/Δep2 lysates. BRAFΔep lysates were characterized by reduced ERK phosphorylation only, while RAF1Δep lysates showed increased phosphorylation of all three MAPK. The expression of ICAM1 as marker of inflammation was detected only in the Δ/Δep2 lysates, leading to the hypothesis that the combination of high JNK, low ERK activation is at the basis of the inflammatory phenotype of Δ/Δep2 animals (*Figure 6C*).

## Inhibition of ERK activation induces inflammatory skin disease in RAF1Δep mice

We tested this hypothesis by treating 6 weeks old RAF1Δep animals with a MEK inhibitor (MEKi; trametinib, in clinical use; daily by gavage for 32 days) to decrease ERK activation and determine whether this would phenocopy the Δ/Δep2 disease. MEKi efficiently reduced ERK phosphorylation and slightly increased JNK phosphorylation in both control and RAF1Δep epidermal lysates, but increased p38 phosphorylation was only observed in the RAF1Δep (*Figure 7A*). Consistently, MEKi reduced the expression of the phosphatases DUSP1 and 10 and of CLDN1 in RAF1Δep epidermis only (*Figure 7A*). Within a month, the RAF1Δep animals developed an inflammatory skin disease characterized by K6 and ICAM1 expression in the epidermis, by the presence of activated dermal mast cells and by an increase in TSLP (*Figure 7B*). In control animals, inhibitor treatment did not affect, or reduced chemo- or cytokine amounts; GMCSF, IL5, IL2 and CCL4 were reduced significantly, and a trend could be observed for IL4, IL17, IL27, and CXCL1. In contrast, a comparison between inhibitor-treated RAF1F/F and RAF1Δep lysates revealed a significant upregulation of CCL2, GMCSF, IL18, IL5, IL13, IL2, IL17 and CCL4 (*Figure 7C* and *Figure 7—figure supplement 1*). This correlated with increased Mac1+Gr1+ cells in the spleen and activated T, B, and dendritic cells in the lymph nodes (*Figure 7D*) as well as with increased circulating IgEs and granulocytosis (*Figure 7E*). Thus, inhibiting ERK in the RAF1Δep animals recreates the immunological environment necessary to bring about a disease very similar to that observed in Δ/Δep2 mice.

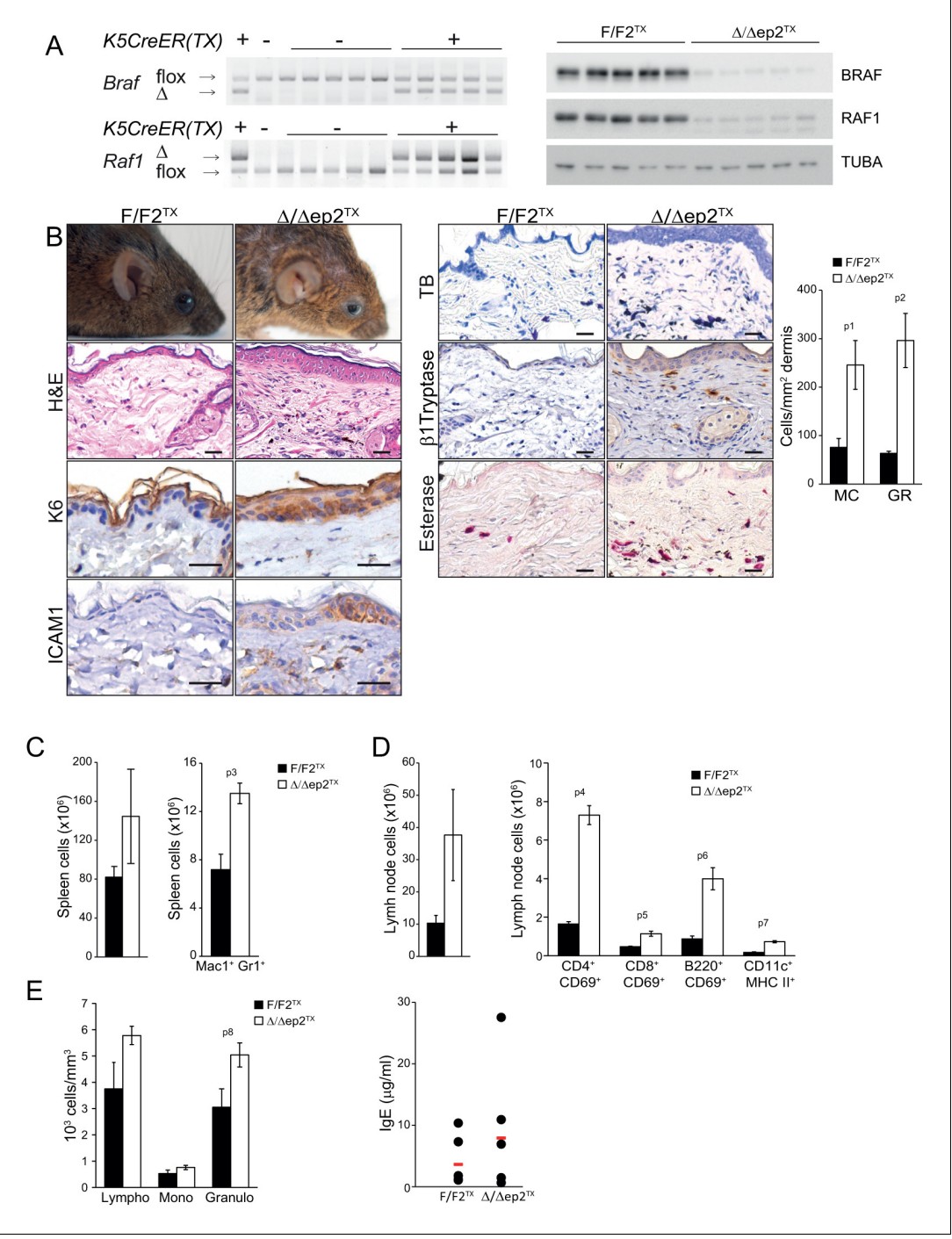

**Figure 5.** Local and systemic inflammation in Δ/Δep2$^{TX}$ mice. (**A**) PCR analysis of tail tissue (left) and immunoblot analysis of epidermal lysates obtained from Δ/Δep2$^{TX}$ animals. (**B**) Macroscopic appearance of Δ/Δep2$^{TX}$ mice and histological analysis of H&E sections. Scale bars, 25 μm. Infiltrating cells: mast cells (MC; TB$^+$), activated mast cells (β1 Tryptase$^+$; modest), granulocytes (GR; esterase$^+$). The plot shows a quantification of the histological analysis. (**C**) Mild splenomegaly with increased numbers of Mac1$^+$Gr1$^+$ cells in Δ/Δep2$^{TX}$ animals. (**D**) Activated T cells, B cells and dendritic cells in the lymph nodes of Δ/Δep2$^{TX}$ animals. (**E**) Mild lymphocytosis and significantly elevated granulocyte numbers in Δ/Δep2$^{TX}$ mice. The right panel shows comparable IgE plasma levels in control and Δ/Δep2$^{TX}$ animals. Data are plotted as mean ± SEM (n = 5; p1 = 0.034, p2 = 0.014, p3 = 0.005, p4 = 2.63E-4, p5 = 0.001, p6 = 0.001, p7 = 0.019 and p8 = 0.042).

*Figure 5 continued on next page*

*Figure 5 continued*

The following figure supplement is available for figure 5:

**Figure supplement 1.** Representative FACS analysis of lymph node and spleen cells isolated from adult F/F2$^{TX}$ Δ/Δep2$^{TX}$ animals.

## The inflammatory skin disease of Δ/Δep2 mice can be prevented by JNK inhibition but not by MyD88, TNF or caspase 1/11 ablation

Collectively, the data above suggest that JNK activation was responsible for the inflammatory skin phenotype. To assess whether this was the case in vivo, we treated Δ/Δep2 animals with the specific peptide inhibitor D-JNKI1. D-JNKI1 efficiently blocked JUN phosphorylation, increased ERK and RSK phosphorylation (*Figure 8A*), and prevented the development of the disease as determined by ICAM1 and TSLP expression in the epidermis as well as by eyelid inflammation and mast cells accumulation in the dermis (*Figure 8A–B*). Consistent with the data in *Figure 3—figure supplement 1B*, K6 expression could not be observed in the interfollicular epidermis in any of the experimental groups (*Figure 8—figure supplement 1A*). Compared to untreated 3 weeks old animals, vehicle (TAT-peptide) alone had minor effects on the level of some chemokines and cytokines; the increase in IL17 and IL18 in TAT-peptide-treated Δ/Δep2 vs F/F2 animals became significant; the opposite was observed for CCL7, IL13 and IL4 (compare *Figures 3B* and *8C*; and *Figure 3—figure supplement 1* with *Figure 8—figure supplement 1B* ). Treatment with D-JNKI1 reduced chemo- and cytokines in epidermal lysates of Δ/Δep2 animals to levels indistinguishable from, or lower than (CCL2, CCL4, GMCSF and IL4), those of F/F2 controls. Exceptions were IL6, which was significantly reduced in Δ/Δep2, but not in F/F2 epidermal lysates; and IL5, which was not affected by the inhibitor in either genotype (*Figure 8C* and *Figure 8—figure supplement 1B*). At the systemic level, D-JNKI1 normalized the numbers of activated B and dendritic cells in lymph nodes, while the numbers of activated T cells where reduced in both F/F2 and Δ/Δep2 lymph nodes (*Figure 8D*). Thus, JNK activation and the resulting chemo- and cytokine accumulation in the epidermis are required for disease development.

To determine whether the increased JNK activation observed in the Δ/Δep2 animals was due to a specific signal, we mated them to MyD88 knockout animals (*Adachi et al., 1998*), to block both TLR (with the exception of TLR3) and IL1 signaling (*Janssens and Beyaert, 2002*). We also used TNFA knockout mice to prevent TNF signaling (*Kuprash et al., 2005*), and caspase 1/11 knockout mice to ablate IL1 and IL18 production (*Smith et al., 1997*). None of these knockouts altered the progression (onset around 2 months of age) or severity of the Δ/Δep2 skin disease (*Figure 8—figure supplement 2*). The Myd88 knockout, however, reduced the splenomegaly observed in the Δ/Δep2 mice. This specific phenotype is caused by an increase in Mac1+/Gr1+ splenocytes (*Figure 2B*), and its selective rescue in the compound Δ/Δep2;MyD88 knockout animals is likely due to the crucial role of MyD88 in the generation of these cells (*Arora et al., 2010*; *Delano et al., 2007*).

## Interfering with the JNK pathway decreases the production of proinflammatory molecules in primary Δ/Δep2 keratinocytes and RAF knockdown HaCat cells

A concomitant increase in JNK activation and decrease in ERK activation is the basis of the skin disease of Δ/Δep2 animals. To test whether this molecular phenotype was cell-autonomous and to gain some insight in the molecular mechanisms underlying JNK activation, we established and analyzed primary keratinocytes cultures. Δ/Δep2 keratinocytes were "primed" for inflammation: they constitutively expressed ICAM1 (*Figure 9A*) and, upon stimulation, significantly increased amounts of chemokines (CCL2, CCL7) and cytokines (IL6, IL18, TSLP, IL13, *Figure 9B*; IL4 was not detected under these conditions). In addition, Δ/Δep2 keratinocytes showed decreased ERK activation in response to EGF and a strong increase in pJNK when co-treated with proinflammatory stimuli (*Figure 9A*). Treatment with D-JNKI1, a cell-penetrating, protease-resistant peptide that prevents JNK interaction with its JBD-dependent targets (*Borsello and Forloni, 2007*), reduced the expression of the inflammatory proteins ICAM1, TSLP, CCL2 and CCL7, but did not have any major effect on ERK activation

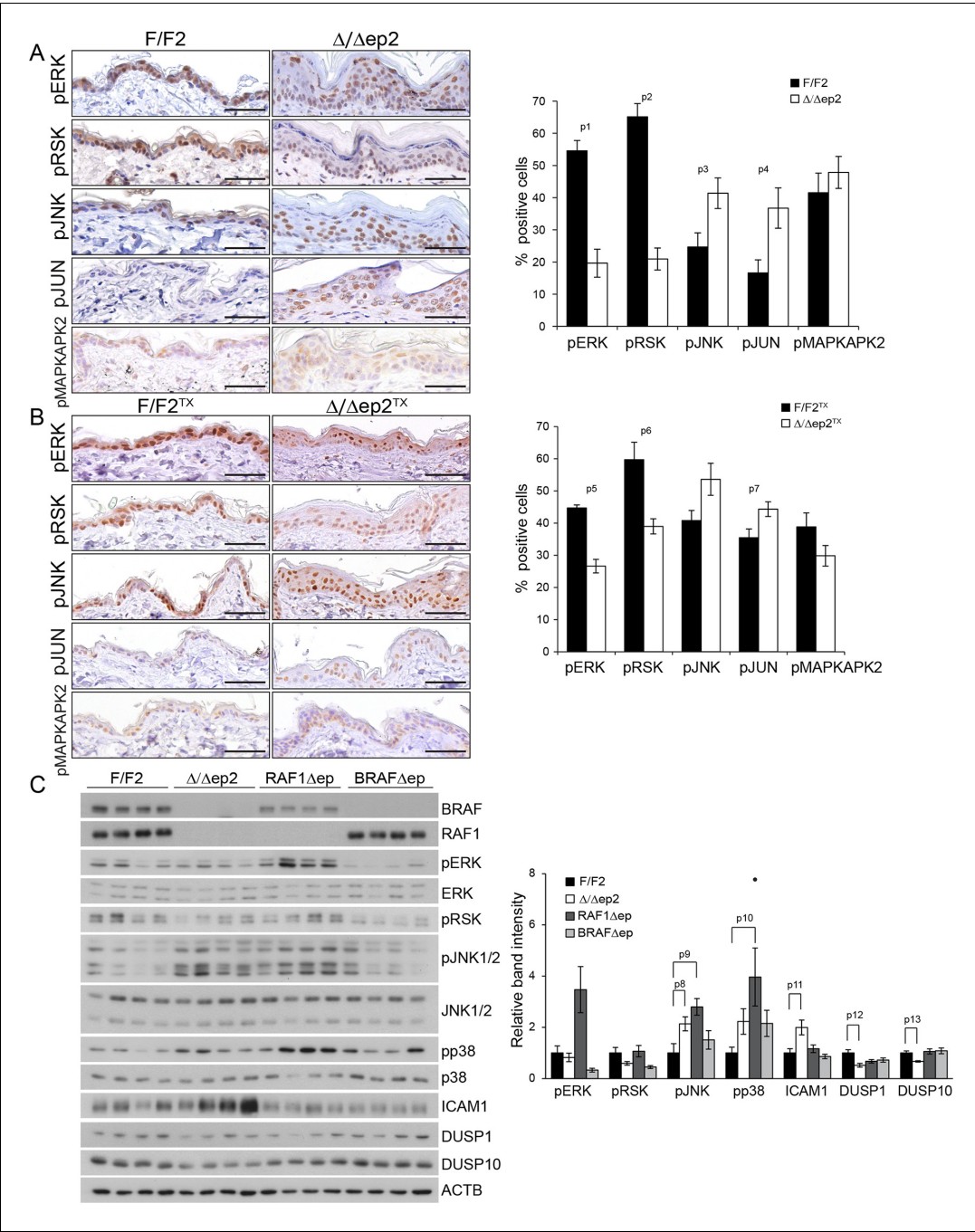

**Figure 6.** Molecular consequences of BRAF/RAF1 deletion in primary keratinocytes and epidermis. (**A, B**) Immunohistochemical analysis of pERK and pJNK, their downstream targets pRSK and pJUN, and the p38 downstream target pMAPKAPK2 in adult F/F2 and Δ/Δep2 (**A**), and F/F2$^{TX}$ Δ/Δep2$^{TX}$ epidermis (**B**). Scale bars, 50 μm. The plots on the right show the percentage of positive cells in the epidermis (n = 4–5). (**C**) Immunoblot analysis of MAPK signaling in 3 weeks old epidermal lysates (n = 4), quantified as in *Figure 1F*. ACTB is shown as a loading control. Phosphorylation is expressed as the ratio between the signals obtained obtained with the phosphospecific antibody and with the protein-specific antibody. In both cases, the data are normalized to one of the F/F2 samples, which was arbitrarily set as 1. Data are plotted as mean ± SEM. p1 = 2.73E-4, p2 = 4.15E-5, p3 = 0.042, p4 = 0.031, p5 = 0.001, p6 = 0.023, p7 = 0.038, p8 = 0.049, p9 = 0.010, p10 = 0.030, p11 = 0.033, p12 = 0.023 and p13 = 0.018.

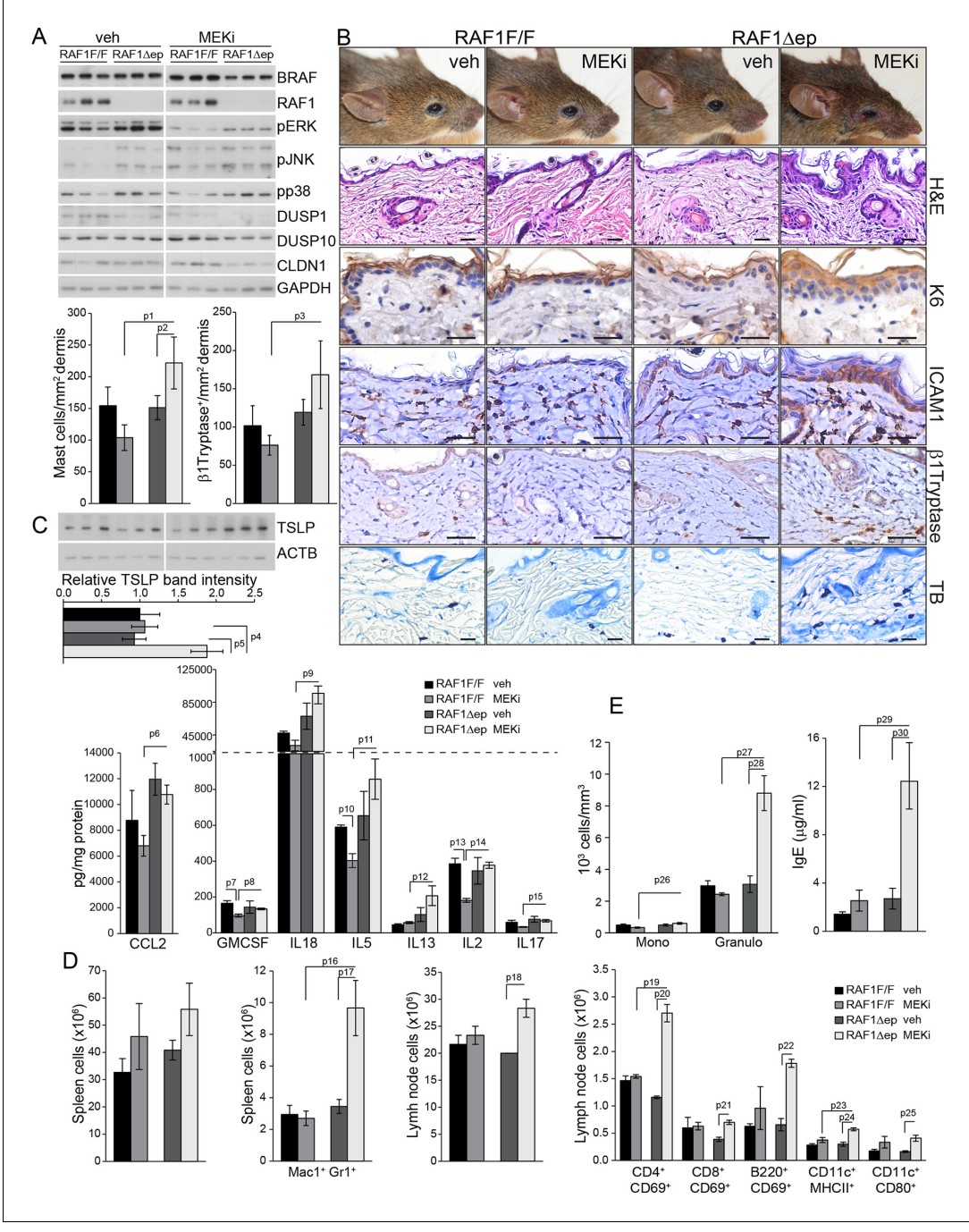

**Figure 7.** MEK/ERK inhibition in RAF1Δep animals phenocopies the Δ/Δep2 phenotype. RAF1Δep animals were treated with a MEK inhibitor (MEKi; trametinib, daily by gavage for 32 days). (**A**) Immunoblot of epidermal lysates showing the effect of MEKi on the phosphorylation and expression of the indicated proteins. GAPDH is shown as a loading control. (**B**) Macroscopic appearance (top panels) and histological analysis of vehicle versus MEKi-treated animals. Mast cells (TB+) and activated mast cells (β1 Tryptase+) are quantified in the plots on the left. Scale bars 25 μm. (**C**) Inflammatory chemokines and cytokines in epidermal lysates of MEKi treated-mice. TSLP levels were determined by immunoblotting and quantified and analyzed as in **Figure 1F**. ACTB served as a loading control. (**D**) Increased numbers of splenic Mac1+ Gr1+ cells and of activated T cells, B cells and dendritic cells in the lymph nodes of MEKi-treated RAF1Δep animals. (**E**) Mild monocytosis and granulocytosis in MEKi-treated RAF1Δep animals and elevated amount of plasma IgE. Data represent mean ± SEM (n = 3; p1 = 0.002, p2 = 0.017, p3 = 0.003, p4 = 0.025, p5 = 0.041, p6 = 0.022, p7 = 0.023, p8 = 0.029, p9 = 0.010, p10 = 0.032, p11 = 0.044, p12 = 0.053, p13 = 0.015, p14 = 0.001, p15 = 0.031, p16 = 0.049, p17 = 0.026, p18 = 0.038, p19 = 0.015,

*Figure 7 continued on next page*

*Figure 7 continued*

p20 = 0.001, p21 = 0.005, p22 = 0.002, p23 = 0.033, p24 = 0.006, p25 = 0.039, p26 = 0.025, p27 = 0.004, p28 = 0.020, p29 = 0.027 and p30 = 0.028).

The following figure supplement is available for figure 7:

**Figure supplement 1.** Epidermal chemokine and cytokine levels in MEKi treated mice.

and on the stress kinase phosphatases DUSP1 and DUSP10, expressed at lower levels in Δ/Δep2 keratinocytes (*Figure 9C–D*).

To ensure that the effects observed in the Δ/Δep2 cells and tissue could be reproduced by acute ablation, we performed knockdown experiments in the human keratinocyte cell line HaCat. Concomitant silencing of *BRAF* and *RAF1* (double knockdown, KD2) abolished basal ERK phosphorylation, decreased ERK activation and increased JNK activation by a combination of EGF and proinflammatory stimuli (*Figure 9—figure supplement 1A*). Constitutive ICAM1 expression was not observed in KD2 cells, but they expressed higher levels of this molecule when treated with TNFα. ICAM1 expression was reduced by D-JNKI1 in KD2 cells and increased by MEKi in RAF1KD cells (*Figure 9—figure supplement 1B–C*). *CCL2* mRNA accumulation, which depends on concomitant stress kinase activation and ERK inhibition, was also elevated in a JNK-dependent manner in KD2 cells (*Figure 9—figure supplement 1B*); MEK inhibition strongly increased *CCL2* mRNA in WT cells (*Pastore et al., 2005*) and even more so in RAF1KD cells (*Figure 9—figure supplement 1C*).

Mixed lineage kinase 3 (MLK3) activates the JNK pathway (*Gallo and Johnson, 2002*) and can function both as a positive regulator of ERK signaling, via kinase-independent mechanisms (*Chadee and Kyriakis, 2004*; *Chadee et al., 2006*) and as a negative regulator of ERK, via JNK-dependent mechanisms (*Shen et al., 2003*). Importantly, MLK3 binds to JIP1 (*Whitmarsh et al., 1998*), and signal flow from this upstream kinase would be interrupted by D-JKNI1. We thus determined whether MLK3 was implicated in the imbalance in ERK/JNK signaling observed in Δ/Δep2 primary keratinocytes and in KD2 HaCat cells. MLK3 downregulation by two independent shRNAs strongly reduced both ERK and JNK phosphorylation in F/F2 cells treated with EGF and proinflammatory stimuli; however, in Δ/Δep2 cells, only JNK phosphorylation was reduced (*Figure 9E*). Consistently, MLK3 downregulation reduced ICAM1 upregulation and the induction of *Ccl2* and *Tslp* mRNA (*Figure 9F*). Essentially the same results were obtained by downregulating MLK3 in WT and KD2 HaCat cells (*Figure 9—figure supplement 1D*). Thus, MLK3 is responsible for JNK activation and the induction of inflammatory molecules in Δ/Δep2 keratinocytes and HaCat cells.

## Discussion

### BRAF and RAF1 prevent keratinocyte-driven allergic inflammation

Keratinocytes have long been known to contribute to the pathogenesis of skin inflammatory disorders, but whether they do so by reacting to stimuli produced by immune cells or by initiating the cascade leading to the disease is still a matter of debate. We describe an essential role of BRAF and RAF1 in keratinocytes in the control of allergic inflammation. Mice lacking BRAF/RAF1 in keratinocytes develop a disease clinically very similar to human atopic dermatitis (*Bieber, 2010*). It starts with a barrier defect accompanied by the reduced expression of crucial tight junction proteins and by a marginally increased expression of chemokines and cytokines, particularly CCL2, IL6 and IL18. It then progresses to a stage in which the skin appears unaffected, yet both local (mast cell infiltration, chemokine and cytokine production) and systemic (T and B cell activation, increased amounts of circulating leukocytes and chemokines) anomalies become evident while TSLP and IgE levels are only slightly increased. This is reminiscent of the early phase of atopic dermatitis in children, prior to IgE sensitization (*Bieber, 2010*). Unlike the majority of atopic dermatitis patients (*De Benedetto et al., 2012*), however, the Δ/Δep2 animals don't have a stratum corneum defect, which may help explain the lack of symptoms at this stage. Adult mice present with a full-fledged disease characterized by rich dermal infiltrates, increased cytokine production, clearly elevated IgEs and mast cell activation, in good correlation with the intense pruritus causing extensive scratching and self-inflicted wounds.

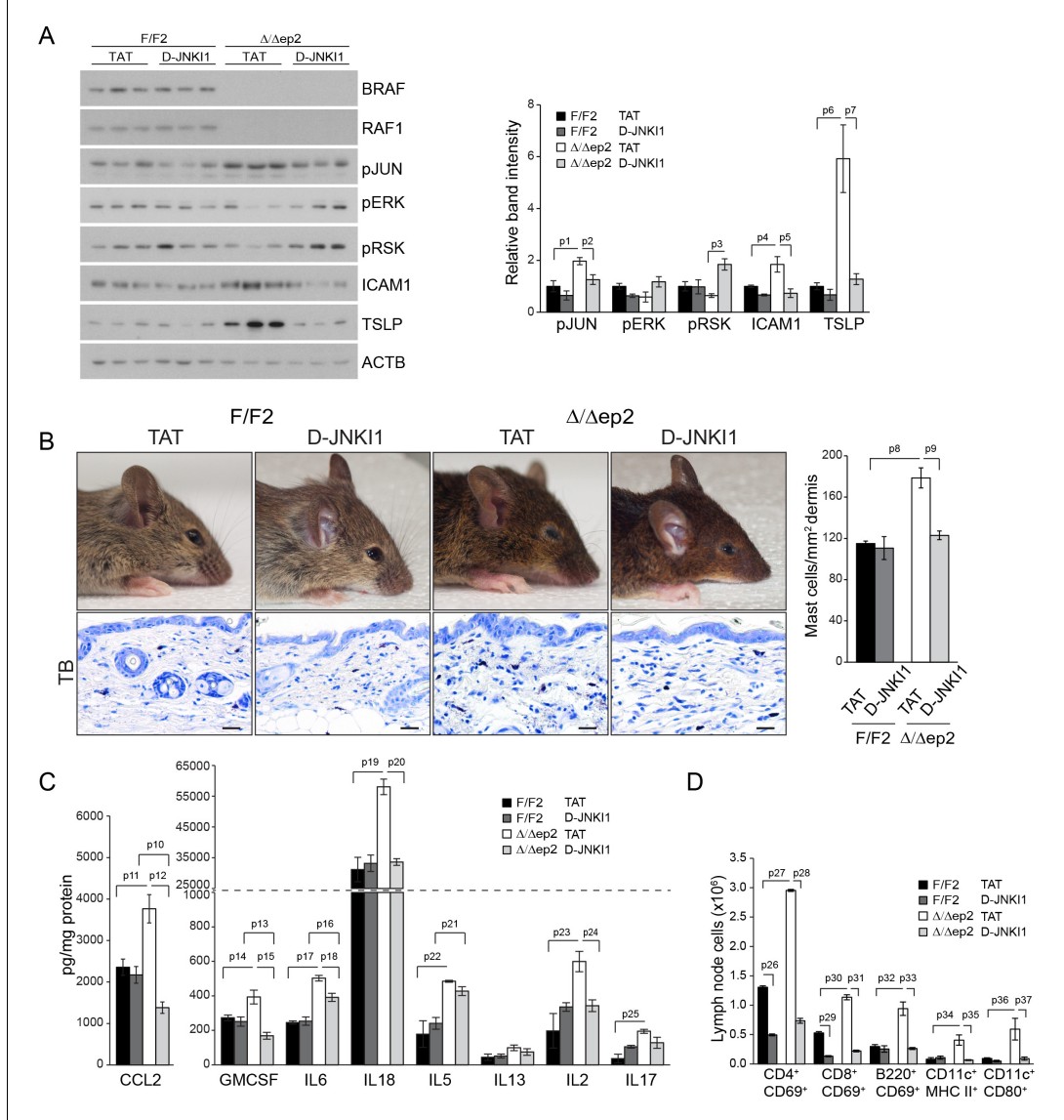

**Figure 8.** D-JNKI1 treatment rescues inflammation in Δ/Δep2 mice. Mice were treated with D-JNKI1 or TAT peptide (22 mg/kg i.p. at 10 days of age) and analyzed after 12 days (**A**) D-JNKI1 treatment prevents disease onset in Δ/Δep2 mice. Immunoblot of epidermal lysates showing the effect of D-JNKI1 on the phosphorylation and expression of the indicated proteins, quantified as in *Figure 1F*. ACTB is shown as a loading control. (**B–D**) Decreased eyelid inflammation, mast cells infiltration (**B**; TB$^+$; quantified in the plot on the right), epidermal chemokine/cytokine levels (**C**) and activated T cells, B cells and dendritic cells in lymph nodes (**D**) in D-JNKI1-treated Δ/Δep2 mice. Scale bars, 25 μm. Data represent mean ± SEM (n = 3–5; p1 = 0.026, p2 = 0.042, p3 = 0.022, p4 = 0.048, p5 = 0.044, p6 = 0.020, p7 = 0.025, p8 = 0.018, p9 = 0.016, p10 = 0.014, p11 = 0.023, p12 = 0.011, p13 = 0.039, p14 = 0.049, p15 = 0.015, p16 = 0.003, p17 = 1.70E-4, p18 = 0.008, p19 = 0.008, p20 = 0.004, p21 = 0.003, p22 = 0.017, p23 = 0.026, p24 = 0.027, p25 = 0.005, p26 = 2.13E-6, p27 = 4.50E-8, p28 = 1.39E-5, p29 = 0.001, p30 = 0.001, p31 = 0.001, p32 = 0.023, p33 = 2.35E-4, p34 = 0.050, p35 = 0.002, p36 = 0.050 and p37 = 0.012).

The following figure supplements are available for figure 8:

**Figure supplement 1.** K6 expression and epidermal chemokine and cytokine levels in D-JNKI1-treated mice.

**Figure supplement 2.** The inflammatory phenotype of Δ/Δep2 mice is not rescued by MyD88, caspase 1/11, or TNF knockout.

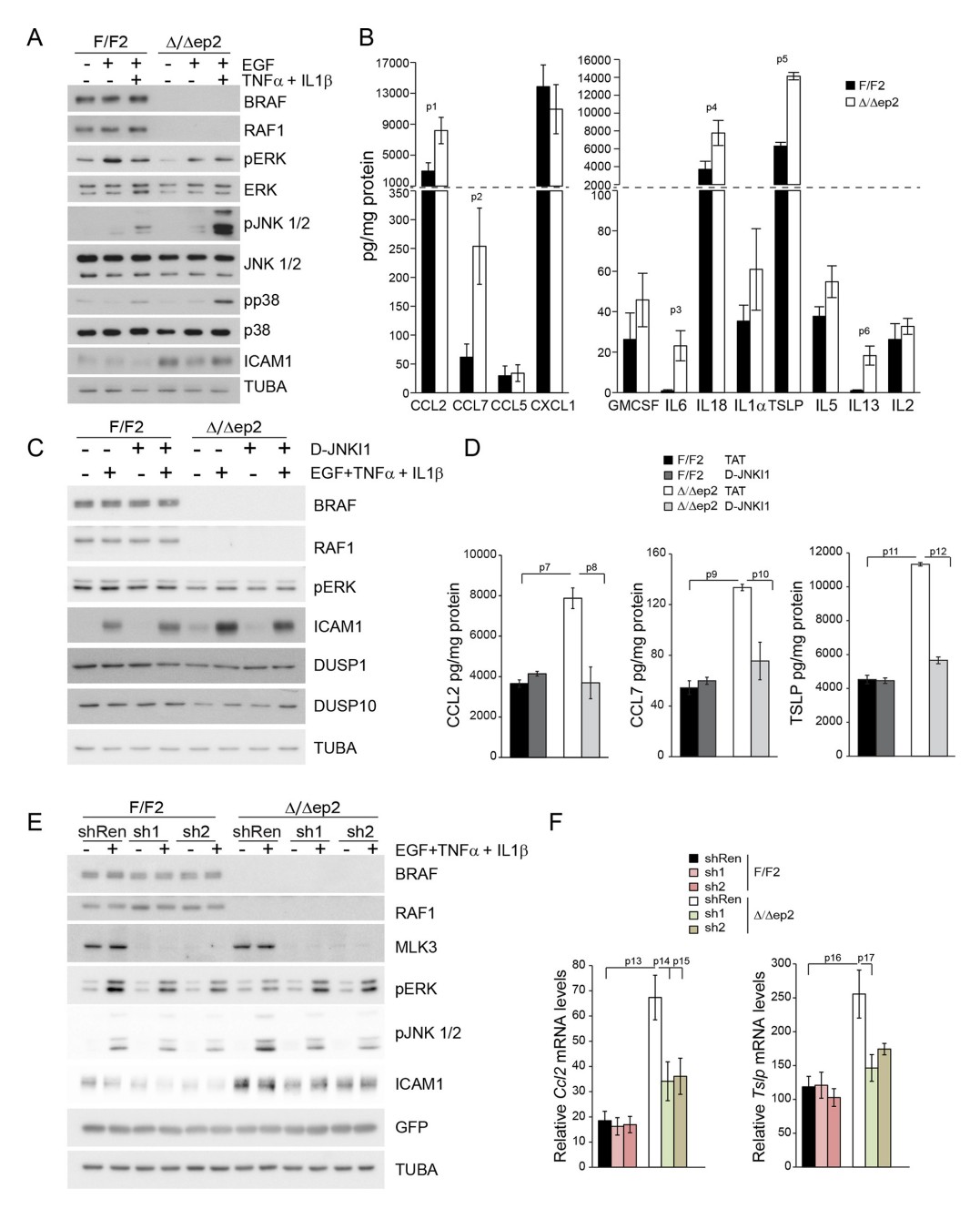

**Figure 9.** Increased stress kinase signaling and JNK pathway-dependent cytokine and chemokine production by primary keratinocytes lacking BRAF and RAF1. (**A**) Reduced ERK phosphorylation and increased JNK/p38 activation in primary Δ/Δep2 keratinocytes stimulated with EGF and/or TNFα and IL1β for 15 min. (**B**) Increased cytokine and chemokine production in primary Δ/Δep2 keratinocytes treated with EGF, TNFα and IL1β for 24 hr. Cytokine and chemokine production was determined by multiplex analysis, except for TSLP which was quantified by ELISA. Data represent mean ± SEM of 3–5 biological replicates. (**C–D**) Cells were pretreated with D-JNKI1 inhibitors prior to stimulation with EGF, TNFα and IL1β for 15 min (**C**) or 24 hr (**D**). Data represent the mean ± SEM of technical replicates (n = 3). (**E–F**) Effect of shRNA-mediated *Mlk3* silencing on ERK and JNK phosphorylation and ICAM1 expression (**E**; stimulation with EGF, TNFα and IL1β for 15 min) and on the expression of *Ccl2* and *Tslp* mRNA (**F**; stimulation with EGF, TNFα and IL1β for 24 hr) by F/F2 and Δ/Δep2 keratinocytes. shRen, shRNA targeting Renilla, used as a control; sh1 and sh2, targeting *Mlk3*, binding sites nucleotide 2266–2285 and 2383–2402, respectively. The shRNAs were encoded by lentiviral vectors coexpressing GFP. GFP immunoblots are shown to confirm similar levels of infection in all samples. Data represent mean ± SEM of 4 biological replicates. Each keratinocyte culture represents a pool of three mice. Immunoblots are representative of three independent experiments. p1 = 0.041, p2 = 0.040, p3 = 1.89E-4, p4 = 0.018, p5 = 0.046, p6 = 0.020, p7 = 0.008, p8 = 0.016, p9 = 0.001, p10 = 0.018, p11 = 3.23E-4, p12 = 1.47E-4, p13 = 0.007, p14 = 0.03, p15 = 0.035, p16 = 0.023 and p17 = 0.046.

*Figure 9 continued on next page*

*Figure 9 continued*

The following figure supplement is available for figure 9:

**Figure supplement 1.** Compound knockdown (KD2) of *BRAF* and *RAF1* induce the expression of inflammation markers by HaCat cells in a MLK3/JNK-dependent manner.

The initial barrier defect contributes to full-fledged disease, since ablation of BRAF/RAF1 after the third week causes a less severe condition lacking IgE sensitization and mast cell activation, yet recapitulating the systemic symptoms of the disease. This could be likened to late-onset, IgE-sensitization-independent dermatitis (*Bieber, 2010*). Thus, BRAF and RAF1 act together to ensure the timely establishment of the inside-out barrier of the epidermis and to prevent allergic inflammation. The data support the hypothesis that the keratinocytes play a primary role in the pathogenesis of atopic dermatitis, and the Δ/Δep2 animals may be useful as a model for this disease (*Figure 10*).

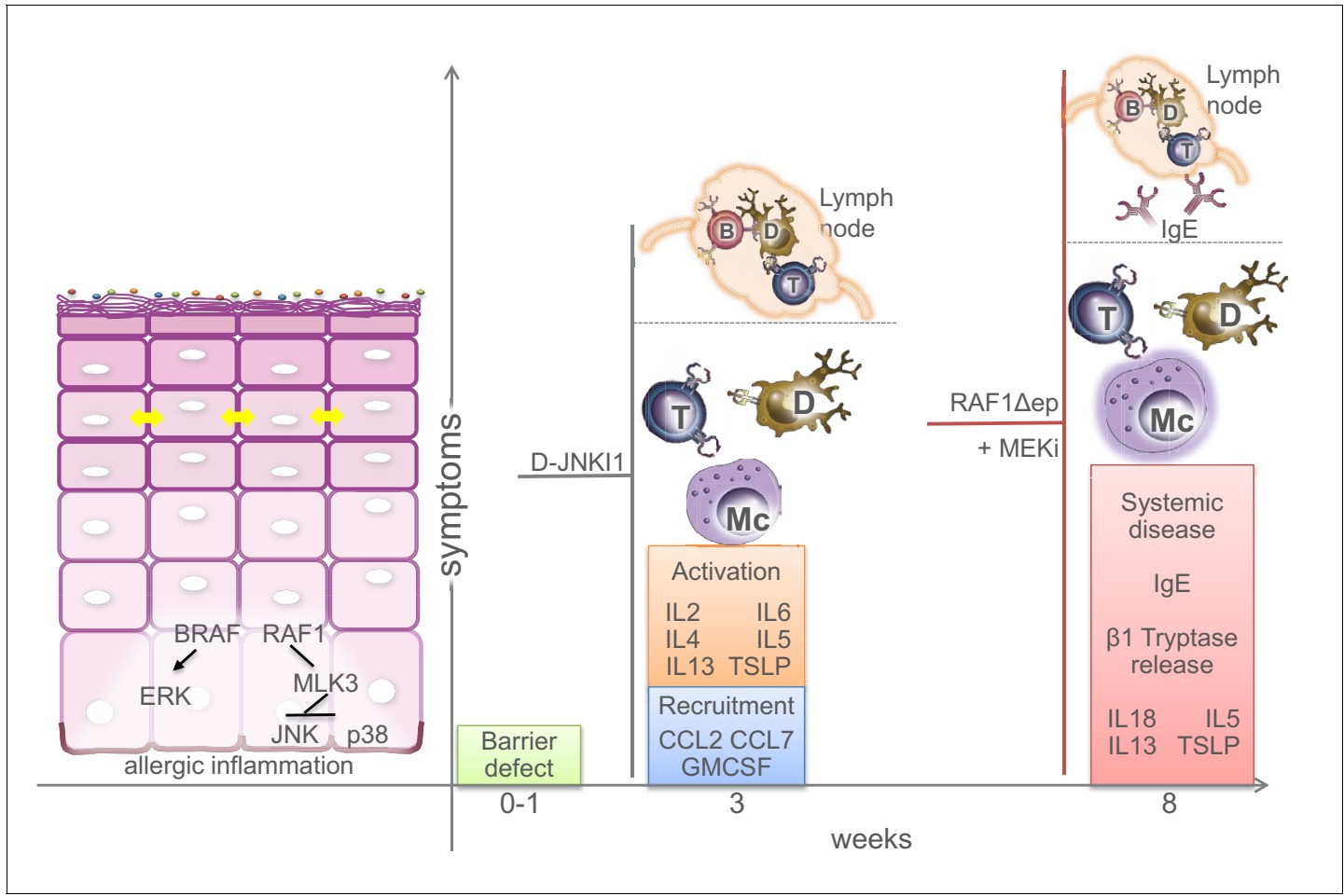

**Figure 10.** Molecular and physiological defects in mice lacking BRAF and RAF1 in the epidermis. Compound BRAF/RAF1 ablation in keratinocytes induces an imbalance in MAPK signaling, resulting in low ERK, high JNK activation. This causes early inside-outside barrier defects accompanied by reduced CDN1 expression (yellow arrows), followed by a breakdown of the immunological barrier and local as well as systemic allergic inflammation akin to atopic dermatitis, characterized by the presence of Th2 cytokines in the epidermis. The phenotype can be prevented by inhibiting the JNK pathway in Δ/Δep2 animals and cells, and phenocopied by inhibiting the ERK pathway in Raf1Δ/Δep animals. Systemic effects (lymph node involvement, circulating IgEs) are separated from local effect by a dashed line. B (B cells), T (T cells), Mc (Mast cells), D (dendritic cells).

## A set of chemokines and Th2 cytokines underlies Δ/Δep2 allergic inflammation

In this respect, our experiments provide information about the minimal set of keratinocyte-derived factors required for the establishment of allergic, atopic dermatitis-like inflammation.

In the phase before IgE sensitization and mast cell activation, we detected significantly higher levels of TSLP, CCL2, CCL7, GMCSF, IL2, IL5, IL6, and IL13 in Δ/Δep2 epidermal lysates; most chemokines and cytokines in the Δ/Δep2 animals were sensitive to D-JNKI1 inhibitor treatment, which brought Δ/Δep2 and F/F2 epidermal lysates down to indistinguishable levels. Conversely, the set of cytokines (TSLP, IL5, IL13, IL18, borderline IL2 and IL17) observed in adult, diseased Δ/Δep2 mice was similar to that promoting symptomatic disease in RAF1Δep animals treated with trametinib, which showed an increase in TSLP, IL2, IL5, IL13, as well as in IL18. These data can be interpreted to mean that ERK signaling negatively regulates the production of allergy-promoting factors. However, it is important to note that trametinib treatment reduced IL2 and IL5 as well as GMCSF in the F/F2 epidermis, indicating that sensitivity to ERK inhibition must be seen in the context of altered cell-autonomous signaling (increased JNK activation in RAF1Δep epidermis) and/or tissue milieu. Along the same lines, Δ/Δep2 keratinocytes treated with inflammatory stimuli were significantly more efficient than F/F2 in producing CCL2, CCL7, IL6, IL18, TSLP and IL13, but not IL2 or IL5, resembling, but not fully reproducing, the spectrum observed in epidermal lysates. At this stage, it is impossible to determine whether these discrepancies might be due to the presence of cell types other than keratinocytes in the epidermal lysates or whether the culture and stimulation conditions used did not recapitulate the situation in vivo.

Be that as it may, the range of chemokines and cytokines overrepresented in the Δ/Δep2 mice and in the trametinib-treated RAF1Δep mice is consistent with allergic inflammation. In this respect, strong expression of CCL2 has been reported in the basal keratinocytes of atopic dermatitis patients (*Giustizieri et al., 2001*), and expression of CCL7, which can attract basophils, eosinophils, mast cells and Th2 cells, can be induced by allergens in atopic skin (*Ying et al., 1995*).

TSLP, the signature cytokine of allergic disease, promotes the functions of these same cell types (*Ziegler et al., 2013*), and is responsible for the extreme itching accompanying atopic dermatitis (*Wilson et al., 2013*). High levels of TSLP (*Liu, 2006*), but also of IL5 and IL13 have been reported in the skin of atopic dermatitis patients, particularly in those with elevated IgEs (*Jeong et al., 2003*). In mice, it could be shown that the expression of TSLP (*Liu, 2006*), IL4, IL5 and IL13 induces atopic dermatitis as well as asthma (*Lee and Flavell, 2004*), and IL5 knockout mice exposed to allergens are less prone than wild-type to develop skin eosinophilia and epidermal thickening (*Spergel et al., 1999*). Finally, expression of IL13 in keratinocytes is sufficient to induce a disease mimicking atopic dermatitis (*Zheng et al., 2009*). Thus, these cytokines appear instrumental for the development of atopic dermatitis with IgE sensitization. IL18, on the other hand, can stimulate both Th1 and Th2 responses (*Nakanishi et al., 2001*), and appears to be causally involved in a different type of atopic dermatitis. IL18 is essential for the induction of dermatitis by the epidermis-restricted expression of caspase 1, the enzyme that generates active IL18 from its precursor. In addition, expression of IL18 in keratinocytes induces a late-onset dermatitis associated with mastocytosis, but independent of IgE production (*Konishi et al., 2002*). More recently, IL18 knockout mice were shown to be resistant to infection-associated atopic dermatitis, in a mouse model generated by perturbing the stratum corneum with detergent prior to topical application of *S. aureus* protein A (*Terada et al., 2006*). However, stratum corneum defects could not be observed in Δ/Δep2 mice, and caspase 1 ablation does not modify the course of the Δ/Δep2 disease. Thus, IL18 is not a determining factor in the context of the Δ/Δep2 dermatitis, which rather resembles a Th2/IgE-driven disease.

## An imbalance in MAPK signaling underlies skin inflammation in Δ/Δep2 mice

Ablation of BRAF and RAF1 in the epidermis has non-redundant functions converging on MAPK cascades. Decreased ERK and increased JNK activation are observed in Δ/Δep2 epidermis in vivo independently of the severity of the disease as well as in Δ/Δep2 keratinocytes and BRAF/RAF1 knockdown HaCat cells. Thus, the imbalance in MAPK signaling represents an intrinsic defect of cells and tissues lacking BRAF/RAF1. While BRAF ablation, as in many other tissues/organs, is responsible for reduced ERK activation (*Desideri et al., 2015*), RAF1 ablation correlates with a previously

unnoticed increase in stress kinase activation. Consistent with this, we have observed a reduction in the expression of two DUSPs involved in the inactivation of stress kinases, which is strongest in the Δ/Δep2 epidermis. This is in line with the reported positive role of p38 in the induction of DUSP expression and of ERK in its attenuation at multiple levels (*Caunt and Keyse, 2013*; *Taxman et al., 2011*), which would predict that decreased ERK signaling would be needed to reduce DUSP expression and stabilize JNK phosphorylation when p38 is active. Importantly, both DUSPs have been implicated in the production of cytokines by immunocompetent cells and in murine models of inflammatory and autoimmune disease (*Lang et al., 2006*).

We could further identify MLK3 as the kinase responsible for JNK activation as well as for the expression of inflammatory molecules in primary Δ/Δep2 keratinocytes and BRAF/RAF1 knockdown HaCat cells costimulated with EGF, TNFα and IL1β (as a proxy for the alterations observed in the Δ/Δep2 animals). The lack of basal JNK activation in unstimulated cells suggests a cross-talk at the level of BRAF/RAF1 and MLK3 rather than direct MLK3 activation by BRAF/RAF1 ablation. Such a cross-talk was previously reported to occur in the context of ERK signaling, which is positively regulated by MLK3 via the stabilization of the RAF dimer (*Chadee and Kyriakis, 2004*; *Chadee et al., 2006*). It is possible that in turn, BRAF and RAF1 would restrict the involvement of MLK3 in the JNK pathway, at the same time promoting ERK and reducing JNK activation in stimulated cells (*Figure 10*).

While the mechanistic details remain to be fully elucidated, the relevance of the imbalance in MAPK signaling for the development of the atopic dermatitis-like disease in Δ/Δep2 animals was shown in vivo in two complementary ways: 1) by preventing disease onset with the specific peptide inhibitor D-JNKI1; and 2) by inducing disease in healthy RAF1Δep animals with a MEK inhibitor (*Figure 10*). The second set of experiments also clearly shows that the function of BRAF and RAF1 in the epidermis is not redundant. Finally, the combined RAF knockout and inhibitor data help explain why in patients the cutaneous toxicity of RAF inhibitors, which increase ERK activation, is mostly related to hyperproliferation (*Anforth et al., 2013*), while inhibitors of ERK activation promote the onset or exacerbate the course of cutaneous inflammatory reactions involving stress kinase activation (*Curry et al., 2014*).

## Materials and methods

### Animal studies and inhibitor treatment

All strains were maintained on a Sv/129 background. *Krt5-CreBraf* $^{f/f}$*;Raf1*$^{f/f}$ mice were generated for this study by mating *Krt5-Cre;Braf*$^{f/f}$*:: Krt5-Cre;Raf1*$^{f/f}$ animals. These strains, the *Krt5-CreER(TX);Braf* $^{f/f}$*;Raf1*$^{f/f}$ mice as well as their genotyping and the tamoxifen-induced deletion of the *RAF* alleles have been previously described (*Kern et al., 2013*; *Ehrenreiter et al., 2005*). Inducible RAF deletion was performed at the age of three weeks.

*Krt5-CreBraf* $^{f/f}$*;Raf1*$^{f/f}$ were mated with MyD88 (*Adachi et al., 1998*), TNF (*Kuprash et al., 2005*), and caspase 1/11 (*Smith et al., 1997*) knockout mice, all maintained on a C57BL/6 background, to test the contribution of the respective signaling pathways to the *Krt5-CreBraf* $^{f/f}$*;Raf1*$^{f/f}$ phenotype.

In selected experiments, the MEK inhibitor GSK1120212 (trametinib, Selleckchem, Germany) was applied daily by gavage for 32 days (*Doma et al., 2013*). The peptide inhibitor D-JNKI1 was synthesized at the Istituto di Ricerche Farmacologiche 'Mario Negri' (Milano, Italy), as previously described (*Borsello et al., 2003*). Ten days old animals were injected once i.p. (22 mg/kg) with D-JNKI1 or TAT peptide control ((D-Pro1,2) retro-(D-HIV-TAT (aa 48-57), Enzo Life Sciences, NY, USA, BML-EI384-0001). Animal experiments were authorized by the Austrian Ministry of Science, Research and Economy.

### Histology and immunohistochemistry

H&E staining, TUNEL, BrdU incorporation and immunohistochemistry were carried out as described (*Ehrenreiter et al., 2009*) on paraffin or cryostat sections. The following antibodies were used: β1 Tryptase (AF1937) and ICAM1 (AF796) from R&D Systems (Minneapolis, MN); CD11c (14–0114) and MHC II (14–5321) from Affymetrix eBioscience (Santa Clara, CA); K6 (905701), K5 (905501), K10 (905401) and Involucrin (924401) from BioLegend (San Diego, CA); F4/80 (AbDSerotec, UK, MCAP497); CD4 (550280) and CD8 (550281) from BD Biosciences (San Jose, CA); Filaggrin (ab24584), pRSK (ab32413) and pJNK (ab4821) from Abcam (UK); pJUN (9164) and pERK (4376)

from Cell Signaling Technology (Danvers, MA). Granulocytes were visualized using the Naphthol AS-D Chloroacetate (specific esterase) kit (Sigma, 91C-1KT) according to the manufacturer's instruction. Toluidine blue staining for mast cells was carried out as described (*Mascia et al., 2013*). Histology images were acquired with a ZEISS microscope Imager M1 (20x/0.5 or 10x/0.3 Plan-NeoFluar objectives) equipped with ZEISS AxioCamMRc5. Data were analyzed with ZEISS Axiovision Release 4.8.1 software (Carl Zeiss, Germany). Fluorescent images were acquired with ZEISS Axioplan2 microscope (40x objective, Zeiss Plan – NEOFLUAR; Num: ap. 1.3) equipped with Spot Pursuit Camera (Visitron Systems, Germany) and analyzed with VisiView software (Visitron Systems).

## Blood analysis and FACS

Peripheral blood cell counts were acquired using V-Sight (Menarini Diagnostics, Italy). Spleen and lymph node cells were stained with antibodies against CD11c (550261), CD4 (553051), MHC II (557000), CD69 (557392), CD80 (553768), CD8 (553032), B220 (553090), Mac1 (553310) and Gr1 (553128) all from BD Bioscience and analyzed by FACSCalibur (BD Bioscience) and FlowJo V10 software (Ashland, OR).

## FlowCytomix analyte assay and ELISA

Cytokines and chemokines were detected in cell supernatants, serum samples and epidermal tissue lysates using the Affymetrix eBioscience bead-based multiplex immunoassay. Data were analyzed with FlowCytomix Pro2.4 software. GCSF (R&D Systems, DY414), TSLP (R&D Systems, DY555) and IgEs (Bethyl Laboratories, E90-115) in serum samples were detected by ELISA according to the manufacturer's protocol.

## Barrier function assays

Water loss assay and toluidine blue dye staining of embryos were carried out as described (*Tunggal et al., 2005*).

## Cell culture

HaCaT cells obtained from the DKFZ and mouse keratinocytes were maintained as described (*Doma et al, 2013*). BRAF (L-003460-00), MLK3 (L-003577-00) and RAF1 (L-003601-00) were silenced using ON-TARGETplus SMARTpool siRNAs (Thermo Fisher Scientific, Waltham, MA). Nontargeting pool (D-001810-10-20) was used as control. In accordance with the supplier's protocol, $5 \times 10^5$ cells were transfected with 25 nM of the previously mentioned oligos. Cells were treated with EGF (2 ng/ml, R&D Systems, 2028-EG) and/or with TNF$\alpha$ (2.5 ng/ml, Millipore, Billerica, MA, 654245) and IL1$\beta$ (2.5 ng/ml, R&D Systems, 401-ML/CF) as indicated. For cytokine/chemokine assays supernatants were collected 24 hr later. In selected experiments, cells were pretreated for 1 hr with medium containing DMSO (for trametinib) or TAT peptide only (for D-JNKI1) or with the following inhibitors: D-JNKI1 (2μM) or trametinib (5μM).

## Lentiviral vectors, cloning and transduction procedure

Two independent shRNAs against mouse MLK3 (shRNA1, binding site 2266–2285 and shRNA2, binding site 2383–2402) were designed as 97-bp oligomers containing a 20bp targeting sequence embedded in a shRNAmir stem, amplified and cloned into Xho and EcoRI sites of the miRE lentiviral recipient vector pRRL.SFFV.GFP.miRE.PGK.Puro (SGEP) (*Fellmann et al., 2013*). The SGEP plasmid containing Renilla shRNA served as a control. Lentiviral vectors were transfected in 293T cells. Viral supernatants were collected after 24 and 48 hr and passed through a 0.45-μm filter (Sarstedt, Germany). Each fresh viral supernatant was used for primary keratinocyte spinfection (1500 *g*, 30 min). Primary keratinocyte cultures were harvested 72 hr after first transduction.

## Immunoblotting

Cell and epidermal lysates prepared as previously described (*Doma et al., 2013*) were immunoblotted using the following primary antibodies (1:1000): TUBA (T9206,) from Sigma; pERK1/2 (9101), ERK1/2 (9102), JNK1/2 (9258), pJNK 1/2 (9251), pJUN (9164), pMAPKAPK2 (3041), pp38 (9211), p38 (9212) and ICAM1 (4915) from Cell Signaling Technology; ACTB (sc-1616), 14-3-3 (sc-1657), BRAF (sc-5284), RAF1 (sc-133) and MLK3 (sc-166639 and sc-536) from Santa Cruz Biotechnology (Dallas,

TX); CDH1 (610181) from BD Biosciences; OCLN (ab31721), pRSK (ab32413), TSLP (ab188766) and DUSP10 (ab140123) from Abcam; GAPDH (ABS16) and DUSP1 (07–535) from Millipore; ICAM1 (AF796, R&D Systems) and CLDN1 (374900, Invitrogen, Karlsbad, CA).

## Quantitative PCR

RNA was isolated using Nucleospin RNA II kit (Macherey-Nagel, Germany). cDNA was prepared using Oligo(dT)$_{18}$ primer, dNTPs, and RevertAidReverse Transcriptase (Thermo Fisher Scientific). qPCR was performed using Go Taq qPCR Master mix (A6002, Promega, Madison, WI). Relative expression was calculated by the ΔΔCT method using *ACTB* as housekeeping gene. Human *CCL2* primers used: for 5'-CAGCCAGATGCAATCAATGC-3' and rev 5'-GCACTGAGATCTTCCTATTGG TGAA-3', human *ACTB* for 5'-AGAGCTACGAGCTGCCTGAC-3' and rev 5'-AGCACTGTGTTGGCG TACAG-3'. Mouse *Ccl2* primers used: for 5'-CCCAATGAGTAGGCTGGAGA-3' and 5'-AAAATGGA TCCACACCTTGC-3'; mouse *Tslp* for 5'-CGACAGCATGGTTCTTCTCA-3' and 5'-CGATTTGC TCGAACTTAGCC-3', mouse *ActB* for 5'-CCTCTATGCCAACACAGTGC-3' and 5'-GTACTCCTGC TTGCTGATCC-3'.

## Statistical analysis

Histological samples from at least three animals per condition and genotype were analyzed by counting or measuring at least 3–5 microscopic field/section and analyzed by ImageJ software. Values are expressed as mean (±SEM). The number of biological replicates and, where applicable, technical replicates are indicated in the figure legends. p values were calculated using the two-tailed Student's t test, hetero- or homoskedastic as determined by a previous F-test of equality of variances or, when indicated, by two-way analysis of variance (ANOVA) test. A p value $\leq 0.05$ is considered statistically significant.

## Acknowledgements

We thank Karin Ehrenreiter, Clemens Bogner, Eszter Doma, Jana Strouhalová, Gijs Versteeg, Sandra Vidak and the animal facility for assistance. We thank Mareike Roth and Johannes Zuber (IMP, Vienna) for their help with the shRNA experiments. The work was supported by grants W1220-B09 and P 19530 (FWF), and INFLA-CARE (European Community), all to MB.

## Additional information

### Funding

| Funder | Grant reference number | Author |
| --- | --- | --- |
| Austrian Science Fund | W1220-B09 | Manuela Baccarini |
| European Commission | INFLA-CARE FP7_HEALTH 223151 | Manuela Baccarini |
| Austrian Science Fund | P 19530 | Manuela Baccarini |

The funders had no role in study design, data collection and interpretation, or the decision to submit the work for publication.

### Author contributions

JR, Conception and design of experiments, Acquisition, analysis and interpretation of data, Preparation of data for publication, Drafting or revising the article; IJ, IF, Acquisition of data, Analysis and interpretation of data, Drafting and revising the article; TN, Conception and design of experiments, Acquisition of data, Analysis and interpretation of data; JDN, Conception and design of the shRNA vectors, Acquisition of data, Analysis and interpretation of data; SEK, Acquisition of data, Analysis and interpretation of data, Drafting the article; CR, Conception and design of inhibitor experiments, Acquisition of data, Analysis and interpretation of data; SB, Synthesized and contributed D-JNKI1, Analysis and interpretation of the D-JNKI1 data, Drafting and revising the article; TB, Analysis and interpretation of D-JNKI1 data, Drafting and revising the article, Contributed unpublished essential

data or reagents; MB, Conception, design and supervision of the project, Analysis and interpretation of data, Drafting and revising the article

## Author ORCIDs

Manuela Baccarini, http://orcid.org/0000-0002-3033-391X

## Ethics

Animal experimentation: The study was performed in accordance with the National Austrian legislation (Law of Animal Experiments 2012 ("TVG-Tierversuchsgesetz"; Federal Law regulating the "Experimentation on living animals" BGBI. I Nr.114/2012) and the overriding EU and international legislation and codes of conduct. Animals were bred and housed in the MFPL facility, which follows the "Charter of Fundamental Rights of the European Union", the opinion of the "European Group on Ethics in Science", and the "Protocol on the Protection and Welfare of Animals". Care of mice is performed by licensed animal caretakers according to FELASA recommendations. Animal experiments involving genetic manipulations are governed by the "Gentechnikgetsetz" (GTG, 12.07.1994). All experiments described in the manuscript were authorized by the Austrian Ministry of Science, Research and Economy (GZ: BMWF-66.006/0030-II/3b/2013 and BMWF-66.006/0012-II/10b/2010).

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
