## [Decision Letter]

Thank you for submitting your work entitled "Epidermal RAF prevents allergic skin disease" for consideration by *eLife*. Your article has been favorably evaluated by Tony Hunter (Senior editor) and four reviewers, one of whom is a member of our Board of Reviewing Editors.

The reviewers have discussed the reviews with one another and the Reviewing Editor has drafted a list of experiments and changes that we feel must be completed before this work could be published in *eLife*. At this point, it would help us to have a response from you concerning the list of requirements and the time it would take to complete these task before we make a final recommendation.

Summary:

The manuscript addresses the role of the RAF kinases BRAF and RAF1 in keratinocytes. Mice with combined loss of expression of these kinases in the epidermis exhibit skin defects similar to the human disease, atopic dermatitis (AD). There is local and systemic inflammation, including increased production of chemokines and TH2-type cytokines. The authors show that BRAF is required for ERK activation and that RAF1 suppresses the activation of the stress-activated MAP kinases JNK and p38. Inhibition of JNK in mice prevents the skin defects while inhibition of MEK in mice lacking epidermal RAF1 promotes these.

The authors conclude that RAF enzymes play a key role in the pathogenesis of the atopic dermatitis-like disease in mice and suggest that this may represent a relevant model for studying the disease. Overall the study is interesting. It builds on previous work addressing the complex interplay between the RAF enzymes and the observations that inhibitors targeting RAF can result in inflammatory skin conditions.

Essential revisions:

1) TSLP is a key cytokine for AD-like disease, thus an increase in TSLP levels is very relevant for the conclusions of this study. However, TSLP staining in the epidermis does not look specific, as it should be localized to the cytoplasm, not to the nucleus. The authors should determine the levels of TSLP by RNA and protein (WB or ELISA) in epidermal extracts in all Figures.

2) It is well accepted that IL-4/IL-13 cytokines play important roles in the development of AD. It is necessary to include analyses of both cytokines in all experiments.

3) The IHC analysis is difficult to interpret; some examples include Figure 3 where the panels are too small; Figure 6—figure supplement 1 where the changes in staining between F/F2 and ∆/∆ep2 are unclear; in Figure 8: please use the same magnification that is used for TSLP staining in all panels to make staining more visible; and in regards to ICAM staining use, a better example that clearly shows the epidermis is needed. In general, the quality of all IHC images should be improved and quantifications should be included.

4) The data on MLK3 expression in HaCat cells are not convincing. Is MLK3 activated by loss of Raf1/Braf? JNK inhibition in KD2 cells in Figure 9 is not clear either, and thus these results should be repeated in ∆/∆ep2 keratinocytes in vitro.

5) There is a large body of evidence presented in the manuscript as data not shown that relates to the investigation of increased JNK activation, such as mating the ∆/∆ep2 mice to MyD88, TNFα and caspase 1/11 knock-out mice. These data should be included in the paper, as these results are relevant to the conclusions of the manuscript.

6) The use of SP600125 is not recommended because it is a relatively non-selective kinase inhibitor. Any experiment where SP600125 was used should be replaced with a more selective inhibitor.

7) The RNAi studies are poorly designed – phenotypes should be confirmed with two independent single oligonucleotides and rescued by expression of an RNAi-resistant cDNA.

8) The authors show that BRAF/RAF1 deletion causes an "inside out" barrier defect (Figure 4). Comparing mice with combined BRAF/RAF1 deletion, with mice presenting single deletion for BRAF or RAF1, would be important. Some studies have shown that RAF1 is implicated in disrupting tight junctions (Oncogene. 2007 26(8):1222-30 PMID: 16924233; J Cell Biol. 2000 148(4):791-800.PMID: 10684259).

9) The transient barrier defect conclusion in E18.5 embryos is unclear. At the end of the subsection “Lack of epidermal BRAF and RAF1 causes transient inside-out barrier defects” it is proposed that this defect affects the progression of the disease. This is partly based on the fact that deletion of Braf/Raf1 at 3 weeks produces a more modest phenotype. This may be one possible explanation. The milder phenotype may be a reflection of less robust gene deletion in the K5CreER(TX) mice. An immunoblot of epidermal lysates for BRAF and RAF1 should be included in Figure 5.

10) Figure 6: First, there appears to be specific activation of one ERK isoform (which one?) based on the pERK immunoblot in Figure 6, whilst in Figure 6 both isoforms are phosphorylated. Second, Figure 6 – significance value for IL-13 missing. Third, Figure 6 – what is the explanation for why D-JNKi specifically suppresses CCL2 and CCL7 levels in the Δ/Δep2 mice, while the JNK/p38 inhibitors reduce levels in both F/F2 and knock-out mice?

11) Figure 7: D-JNKi appears to suppress pERK in F/F2 epidermis to similar levels as found in Δ/Δep2. Is there an explanation for this?

12) Overall the outline of the paper is not very logical and it should be improved for clarity. For example, the analysis of the inflammation markers is often inconsistent, and the cytokines/chemokines shown in Figure 1 should also be shown in the subsequent figures such as Figure 2.

---

## [Author Response]

*Essential revisions:*

1) TSLP is a key cytokine for AD-like disease, thus an increase in TSLP levels is very relevant for the conclusions of this study. However, TSLP staining in the epidermis does not look specific, as it should be localized to the cytoplasm, not to the nucleus. The authors should determine the levels of TSLP by RNA and protein (WB or ELISA) in epidermal extracts in all Figures.

The level of TSLP determined by immunoblotting have been found elevated in adult animals (Figure 1), and in 3 weeks old mice (Figure 3); in addition, TSLP levels are decreased in D-JNKI1-treated ∆/∆ep2 mice (Figure 8) and increased in RAF1∆/∆ep mice treated with MEKi (Figure 7). TSLP amounts, determined by ELISA, are significantly increased in the supernatant of primary Δ/Δep2 keratinocytes; treatment with a D-JNKI1 abolishes this increase (Figure 9); Tslp mRNA levels are also increased in Δ/Δep2 keratinocytes; this increase is abolished by MLK3 silencing (Figure 9). Collectively, the data show that TSLP is elevated in a JNK pathway-dependent manner in Δ/Δep2 tissues and cells.

2) It is well accepted that IL-4/IL-13 cytokines play important roles in the development of AD. It is necessary to include analyses of both cytokines in all experiments.

These data were already included in the manuscript, either in the main Figures (original 1F, adult mice – now Figure 1; Figure 3, 3-weeks old mice; original Figure 7, and Figure 7—figure supplement 1,D-JNKI1 – now Figure 8 and Figure 8—figure supplement 1; original Figure 8 and Figure 8—figure supplement 1, trametinib – now Figure 7 and Figure 7—figure supplement 1; original Figure 6 – now Figure 9, keratinocytes). IL-4 was not detected, this is stated in the first paragraph of the subsection “Interfering with the JNK pathway decreases the production of proinflammatory molecules in primary Δ/Δep2 keratinocytes and RAF knockdown HaCat cells”..

3) The IHC analysis is difficult to interpret; some examples include Figure 3 where the panels are too small; Figure 6—figure supplement 1 where the changes in staining between F/F2 and ∆/∆ep2 are unclear; in Figure 8: please use the same magnification that is used for TSLP staining in all panels to make staining more visible; and in regards to ICAM staining use, a better example that clearly shows the epidermis is needed. In general, the quality of all IHC images should be improved and quantifications should be included.

We used low magnification to show a larger portion of the epidermis; however, we agree with the reviewers that some of the immunostainings are less clear in this magnification. The revised manuscript includes higher magnifications in Figure 3, Figure 3—figure supplement 1, Figure 6—figure supplement 1 (original manuscript; now Figure 6); in Figure 8 (now Figure 7), the same magnification originally used for the TSLP panel has been used for K6 and ICAM1; however, the original lower magnification is still shown for the panels showing changes in the dermis, i.e.: H&E, β1-tryptase, and toluidine blue. Quantifications of the IHC analysis are shown wherever they are possible.

4) The data on MLK3 expression in HaCat cells are not convincing. Is MLK3 activated by loss of Raf1/Braf? JNK inhibition in KD2 cells in Figure 9 is not clear either, and thus these results should be repeated in ∆/∆ep2 keratinocytes in vitro.

We have succeeded in performing this experiment (JNK inhibition by MLK3 silencing) in ∆/∆ep2 keratinocytes, even if knockdown experiments in primary cells are notoriously difficult. The lentiviral vectors used to knockdown MLK3 in keratinocytes were constructed by Joanna Nowacka in the lab. The authors’ list has been revised to reflect her contribution. The new experiment is shown in revised Figure 9, while the original experiment in HaCat cells is now shown in Figure 9—figure supplement 1). Concerning MLK3 activation, we have now tried several different phosphospecific antibodies against the autophosphorylation sites, all of which gave a very high background but no reliable signal. However, the lack of JNK activation in unstimulated Δ/Δep2 keratinocytes suggests a cross-talk at the level of BRAF/RAF1 and MLK3 rather than direct MLK3 activation by BRAF/RAF1 ablation. A sentence to this effect has been added to the Discussion (subsection “An imbalance in MAPK signaling underlies skin inflammation in Δ/Δep2 mice”, second paragraph).

5) There is a large body of evidence presented in the manuscript as data not shown that relates to the investigation of increased JNK activation, such as mating the ∆/∆ep2 mice to MyD88, TNFα and caspase 1/11 knock-out mice. These data should be included in the paper, as these results are relevant to the conclusions of the manuscript.

The revised manuscript includes representative pictures of the animals and of their lymphoid organs (Figure 8—figure supplement 2) as well as information on the number of animals monitored (Legend to Figure 8—figure supplement 2) and on the time of disease onset. The paragraph read as follows:

“None of these knockouts altered the progression (onset around 2 months of age) or severity of the Δ/Δep2 skin disease (Figure 8—figure supplement 2). The Myd88 KO, however, reduced the splenomegaly observed in the Δ/Δep2 mice. This specific phenotype is due to an increase in Mac1+/Gr1+ splenocytes (Figure 2), and its selective rescue in the compound Δ/Δep2;MyD88 KO animals is likely due to the crucial role of MyD88 in the generation of these cells (Arora et al., 2010; Delano et al., 2007).”

*6) The use of SP600125 is not recommended because it is a relatively non-selective kinase inhibitor. Any experiment where SP600125 was used should be replaced with a more selective inhibitor.*

All experiments involving SP600125 have been replaced with experiments in which JNK activation was inhibited by D-JNKI1. In addition, the experiment in revised Figure 9 now clearly shows that D-JNKI1 reduces ICAM1 expression in ∆/∆ep2 primary keratinocytes.

7) The RNAi studies are poorly designed – phenotypes should be confirmed with two independent single oligonucleotides and rescued by expression of an RNAi-resistant cDNA.

The RNAi studies involving knockdown of Raf1 and BRaf are backed up by all the experiments performed in knockout mice and cells, making off-target effects more than unlikely. Therefore, we have attempted to perform the requested experiments/controls only for MLK3 knockdown in Δ/Δep2 keratinocytes (see also response to point 4). Knockdown/re-expression experiments in primary mouse keratinocytes are notoriously difficult; we have succeeded in targeting MLK3 with two separate hairpins encoded by a lentiviral vector, but the reconstitution experiments using a single vector expressing both the hairpin and WT or shRNA-resistant MLK3 have failed because the infection efficiency with these large constructs went from 60-90% to less than 20%. This is a problem because keratinocytes cannot be FACS-sorted – they will not reattach and in fact will die after sorting. Irrespective of this, we now show that MLK3 knockdown by two distinct shRNAs in primary keratinocytes (revised Figure 9 and Figure legend to Figure 9; discussed in the last paragraph of the subsection “Interfering with the JNK pathway decreases the production of proinflammatory molecules in primary Δ/Δep2 keratinocytes and RAF knockdown HaCat cells”) has the same effect as MLK3 knockdown by a pool of siRNAs targeting distinct sites in HaCat cells (Figure 9—figure supplement 1 of the revised version). We are therefore confident that the effects of MLK3 silencing are specific.

8) The authors show that BRAF/RAF1 deletion causes an "inside out" barrier defect (Figure 4). Comparing mice with combined BRAF/RAF1 deletion, with mice presenting single deletion for BRAF or RAF1, would be important. Some studies have shown that RAF1 is implicated in disrupting tight junctions (Oncogene. 2007 26(8):1222-30 PMID: 16924233; J Cell Biol. 2000 148(4):791-800.PMID: 10684259).

This experiment has been included in the revised manuscript (Figure 4—figure supplement 1; discussed in the first paragraph of the subsection “Lack of epidermal BRAF and RAF1 causes transient inside-out barrier defects”). The figure shows that neither single knockout shows inside-out barrier defects or reduced tight junction proteins expression.

9) The transient barrier defect conclusion in E18.5 embryos is unclear. At the end of the subsection “Lack of epidermal BRAF and RAF1 causes transient inside-out barrier defects” it is proposed that this defect affects the progression of the disease. This is partly based on the fact that deletion of Braf/Raf1 at 3 weeks produces a more modest phenotype. This may be one possible explanation. The milder phenotype may be a reflection of less robust gene deletion in the K5CreER(TX) mice. An immunoblot of epidermal lysates for BRAF and RAF1 should be included in Figure 5.

This experiment has been included in Figure 5. As described in the second paragraph of the subsection “Lack of epidermal BRAF and RAF1 causes transient inside-out barrier defects” and shows nearly complete deletion of both proteins in epidermal lysates.

10) Figure 6: First, there appears to be specific activation of one ERK isoform (which one?) based on the pERK immunoblot in Figure 6, whilst in Figure 6 both isoforms are phosphorylated. Second, Figure 6 – significance value for IL-13 missing. Third, Figure 6 – what is the explanation for why D-JNKi specifically suppresses CCL2 and CCL7 levels in the Δ/Δep2 mice, while the JNK/p38 inhibitors reduce levels in both F/F2 and knock-out mice?

In the revised version of the manuscript, Figure 6 (now 9C) has been replaced by a new figure showing the effect of the specific JNK inhibitor D-JNKI1. We agree with the reviewer, however, that there is some preference for the phosphorylation of one isoform, and this is ERK2. This can also be observed in epidermal lysates throughout the paper and in HaCat cells. The reason for this preference is not known, but unequal phosphorylation of ERK isoforms has been frequently observed in many different tissues and cells. Figure 6 (now 9B), we stand corrected; this value has been inserted in the revised Figure. Figure 6 (now removed), the explanation is likely that SP600125+SB203580 have a stronger impact and more off target effects than the specific D-JNKI1. In fact we have been requested to remove the experiments in which SP600125 was used for exactly this reason.

11) Figure 7: D-JNKi appears to suppress pERK in F/F2 epidermis to similar levels as found in Δ/Δep2. Is there an explanation for this?

There is no obvious or easy explanation for this result; however, it should be noted that gradual differences in intensity observed in IHC may be difficult to interpret. We have therefore performed immunoblot analysis of epidermal lysates, which has given us a more quantitative measure of the differences observed in Figure 7 (Figure 8 in the revised version) and does not show a significant reduction in pERK or in the downstream target pRSK in F/F2 epidermal lysates.

12) Overall the outline of the paper is not very logical and it should be improved for clarity. For example, the analysis of the inflammation markers is often inconsistent, and the cytokines/chemokines shown in Figure 1 should also be shown in the subsequent figures such as Figure 2.

We have done our best to improve the manuscript’s clarity, describing all the in vivo studies first and moving the experiments with cultured cells, now restricted to keratinocytes, to the end of the results. We have also inverted the order of Figure 7 and Figure 8; in the current version of the manuscript, the in vivo experiments using D-JNKI1 directly precede the experiments in primary keratinocytes, which also explore the impact of the JNK pathway on the phenotype in culture. Concerning inflammation markers, we have added K6 as an inflammation marker in addition to ICAM1 in all experiments. The cytokines/chemokines shown in Figure 1 are shown for each experiment, but in the Figure supplements rather than in the main Figures, which are already extremely crowded.